# **PyVision-RL**: Forging Open Agentic Vision Models via RL

Shitian Zhao [* ‡ 1]   Shaoheng Lin [* 1]   Ming Li [2]   Haoquan Zhang [3]   Wenshuo Peng [4]
Kaipeng Zhang [† 5]   Chen Wei [* † 6]

## Abstract

Reinforcement learning for agentic multimodal models often suffers from interaction collapse, where models learn to reduce tool usage and multi-turn reasoning, limiting the benefits of agentic behavior. We introduce `PyVision`-RL, a reinforcement learning framework for open-weight multimodal models that stabilizes training and sustains interaction. Our approach combines an over-sampling–filtering–ranking rollout strategy with an accumulative tool reward to prevent collapse and encourage multi-turn tool use. Using a unified training pipeline, we develop `PyVision`-Image and `PyVision`-Video for image and video understanding. For video reasoning, `PyVision`-Video employs on-demand context construction, selectively sampling task-relevant frames during reasoning to significantly reduce visual token usage. Experiments show strong performance and improved efficiency, demonstrating that sustained interaction and on-demand visual processing are critical for scalable multimodal agents. Code, data and models are released at https://github.com/agents-x-project/PyVision-RL

## 1. Introduction

Large Language Models (LLMs) have rapidly evolved from passive chatbots into actionable agents capable of multi-turn interaction and tool use. Beyond proprietary systems, a growing body of research has explored how to endow open-weight models with tool-using capabilities, particularly for tasks such as deep research and computer use that require sustained interaction with external environments.

More recently, this agentic paradigm has been extended from purely textual domains to multimodal reasoning. Works such as OpenAI o3 (OpenAI, 2025) demonstrate that incorporating tool use into visual understanding can ground multimodal reasoning in task-relevant visual evidence, enabling models to actively manipulate visual inputs rather than passively consume them. This motivates the development of multimodal agents that reason, act, and interact over images and videos.

Existing approaches to multimodal tool use largely follow two design paradigms. One line of work relies on static toolsets, where a fixed set of task-specific tools, such as cropping, zooming, or video clipping, is manually predefined and exposed to the model (Hu et al., 2024; Yang et al., 2023; Gupta & Kembhavi, 2023; Zhang et al., 2025a; Yang et al., 2025; Gao et al., 2025b; Meng et al., 2025b). While effective for specific tasks, these approaches lack flexibility and require task-dependent engineering. An alternative paradigm, dynamic tooling, treats Python as a primitive tool, allowing the model to synthesize task-specific operations on the fly (Zhao et al., 2025a; Zhang et al., 2025b; Hong et al., 2025; Song et al., 2025; Guo et al., 2025b). This approach enables expressive and compositional tool use, but has so far remained largely limited to image understanding and often relies on proprietary APIs, leaving open-weight multimodal RL underexplored, especially for video.

A key challenge in training such agentic multimodal models lies in training stability and avoding interaction collapse. Prior work observes that after RL fine-tuning, models tend to reduce tool usage, converging to short, low-interaction behaviors (Zhang et al., 2025b; Hong et al., 2025). This has led to skepticism about the effectiveness of test-time interaction scaling for agentic visual understanding, in contrast to its success in textual reasoning (Jaech et al., 2024; Li et al., 2025a). We argue that this limitation does not reflect an inherent weakness of interaction, but rather insufficient training incentives and unstable rollout selection during RL.

In this paper, we present an agentic training framework, `PyVision`-RL, for open-weight multimodal models that addresses these challenges. We adopt Python as a primitive tool to enable dynamic tooling for both image and video understanding, and apply reinforcement learning with two key innovations: (1) an oversam-

*Core Contributor †Corresponding Author ‡Project Lead ¹Shanghai AI Lab ²UMD ³CUHK ⁴THU ⁵Alaya Lab ⁶Rice University. Correspondence to: Chen Wei <cw220@rice.edu>, Kaipeng Zhang <kaipeng.zhang@shanda.com>.

*Proceedings of the 43rd International Conference on Machine Learning*, Seoul, South Korea. PMLR 306, 2026. Copyright 2026 by the author(s).

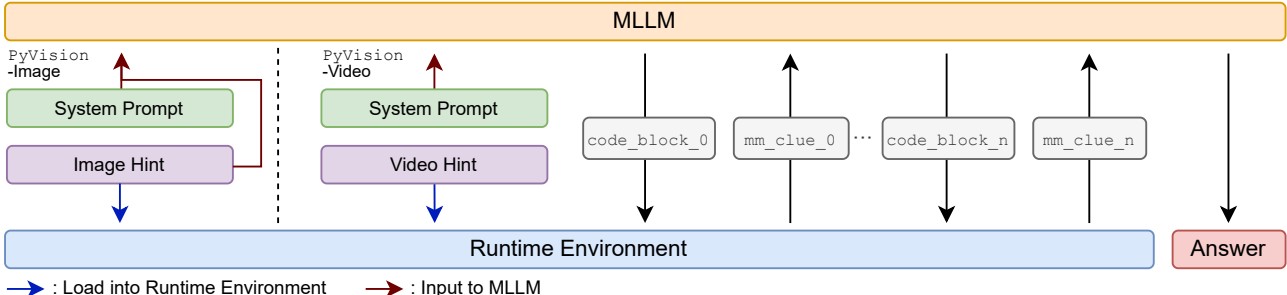

*Figure 1.* **Agentic scaffolds of PyVision-RL.** We design two agentic scaffolds for image and video understanding under a unified framework of dynamic tooling with Python. For PyVision-Image, both the system prompt and image hints are injected into the MLLM context, and the images are also loaded into the Python runtime. For PyVision-Video, only the system prompt is injected into the MLLM context, while the video is loaded exclusively into the runtime environment. Given a query, the model interleaves reasoning with executable code blocks (code_block_0) to process multimodal inputs. Execution results (mm_clue_0), including textual outputs and rendered images, are appended to the context and fed back to the model. This interaction loop repeats until a final answer is produced. By restricting video inputs first to the runtime, PyVision-Video enables on-demand context construction, where the agent selectively samples and plots task-relevant frames during reasoning, substantially improving visual token efficiency (Fig. 2).

pling–filtering–ranking framework for rollout generation that stabilizes agent–environment interaction, and (2) an accumulative tool reward that explicitly incentivizes sustained multi-turn tool usage. Using a unified training pipeline, we introduce two models: PyVision-Image for image understanding and PyVision-Video for video understanding. Especially, PyVision-Video employs on-demand context construction, where the full video is loaded only into the Python runtime, and the model selectively samples and plots task-relevant frames via Python code during the reasoning process. This agentic frame fetching strategy avoids uniform frame sampling, substantially reducing visual token consumption while improving reasoning efficiency.

Our models achieve strong empirical results. PyVision-Image attains state-of-the-art performance on visual search, multimodal reasoning, and agentic reasoning benchmarks, outperforming prior methods such as DeepEyes-v2 (Hong et al., 2025) by +6.9% on V* (Wu & Xie, 2024) and +9.6% on WeMath (Qiao et al., 2025a). PyVision-Video surpasses VITAL (Zhang et al., 2025a), an multimodal agent with a video clipping tool, by +2.2% on VSI-Bench (Yang et al., 2024), while using significantly fewer visual tokens. Enabled by on-demand context construction, PyVision-Video achieves a favorable performance–efficiency trade-off, using on average 5K visual tokens per sample compared to 45K for Qwen2.5-VL-7B, yet attaining higher accuracy: 44.0% for PyVision-Video, 38.0% for Qwen2.5-VL-7B.

In summary, we present PyVision-RL, a unified agentic reinforcement learning framework for open-weight multimodal models that enables tool-based reasoning over both images and videos. By combining an oversampling–filtering–ranking rollout strategy and an accumulative tool reward, our approach prevents interaction collapse and effectively incentivizes multi-turn agent behavior.

The resulting models, PyVision-Image and PyVision-Video, demonstrate that sustained interaction and tool use remain powerful mechanisms for multimodal reasoning when trained with appropriate incentives, achieving state-of-the-art performance while substantially improving token efficiency, particularly for video understanding.

## 2. Related Work

**Tool-Integrated Multimodal Reasoning.** Unlike multimodal reasoning models that rely solely on textual reasoning (Wang et al., 2025a; Deng et al., 2025; Xie et al., 2025), tool-integrated multimodal reasoning explicitly incorporates tool invocation and executed visual outputs into the reasoning process (Wang et al., 2024c). For instance, when analyzing high-resolution images, models may crop or zoom into regions of interest to improve understanding.

Existing approaches broadly fall into two categories. Static toolsets predefine a fixed set of task-specific tools. For visual search, models are equipped with hand-designed cropping and zooming operations specified in the system prompt (Zheng et al., 2025c; Lai et al., 2025; Su et al., 2025a; Hu et al., 2024; Surís et al., 2023; Gupta & Kembhavi, 2023; Song et al., 2026). Similar designs extend to long-video reasoning, where predefined video clipping tools are used (Zhang et al., 2025a; Yang et al., 2025; Gao et al., 2025b; Meng et al., 2025b). In contrast, dynamic tooling treats Python as a primitive tool, allowing models to implement task-specific operations on the fly (Zhao et al., 2025a; Zhang et al., 2025b; Hou et al., 2025; Song et al., 2025; Guo et al., 2025b; Hong et al., 2025). While this paradigm has shown strong results for image tasks, it has not yet been applied to video reasoning. Our method, PyVision-RL, adopt Python as primitive tool, enabling dynamic tooling for image and video understanding tasks, respectively.

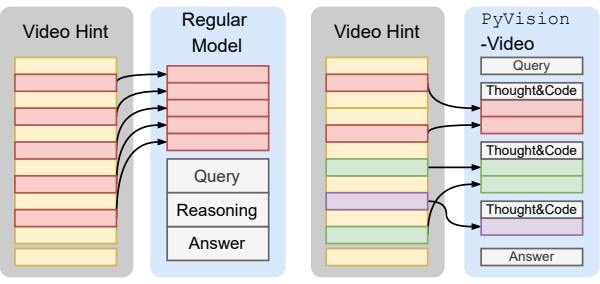

(a) sampling frames     (b) on demand context construction

*Figure 2.* **Comparison between frame sampling and on-demand context construction.** (a) Conventional video MLLMs, *e.g.*, the Qwen-VL series, process videos by uniformly sampling frames and directly injecting them into the model context. (b) In `PyVision`-Video, we adopt on-demand context construction: the video is loaded only into the Python runtime, and the model selectively samples and plots relevant frames via Python code during the reasoning process, largely improve the token efficiency.

**RL for Multimodal Large Language Models.** Following the success of DeepSeek-R1 (Guo et al., 2025a), a growing body of work has applied reinforcement learning to enhance the reasoning and tool-use capabilities of LLMs and multimodal LLMs (MLLMs) (Meng et al., 2025a; Yu et al., 2025; Zheng et al., 2025a). Most of these approaches adopt critic-free RL algorithms.

Existing methods can be broadly categorized by their technical focus. Several works propose improved advantage estimation schemes (Liu et al., 2025c; Hu, 2025). Others modify the PPO-style clipping mechanism to better accommodate LLM training (Yu et al., 2025; Chen et al., 2025a; Zheng et al., 2025b; Zhao et al., 2025b; Gao et al., 2025a). Another line of work addresses training–inference mismatch in RL pipelines (Yao et al.; Liu et al., 2025b), while recent studies focus on stabilizing RL training for large mixture-of-experts (MoE) models (Ma et al., 2025; Xiao et al., 2026).

## 3. Method: `PyVision-RL`

This section introduces `PyVision`-RL, our agentic reinforcement learning framework for training open-weight multimodal models with dynamic tool use. `PyVision`-RL adopts Python as a primitive tool and couples it with a unified agentic scaffold that supports both image and video understanding. The framework is designed to prevent interaction collapse during reinforcement learning and to enable efficient multimodal reasoning. We first describe the agentic scaffold and interaction protocol, then present our RL formulation and training strategies that improve rollout quality and sustain multi-turn tool usage.

### 3.1. Agentic Scaffold: Python as a Primitive Tool

**Interaction Protocol.** As illustrated in Fig. 1, the MLLM is prompted to interleave natural language reasoning with executable code. Specifically, the model generates reasoning text and code blocks `code_block_i`, which are wrapped in `...` tags. The environment executes each code block and returns the execution result `mm_clue_i`, wrapped in `<interpreter> ...</interpreter>` tags. This interaction loop continues until the model produces a final answer, wrapped in `<answer>...</answer>`. All intermediate reasoning, code, and execution outputs are appended to the context.

**Multimodal Hint Injection.** For multimodal understanding tasks such as image and video QA, multimodal hints (images or videos) must be injected into both the MLLM context and the Python execution environment. We adopt separate designs for image and video inputs.

For image tasks, we inject the image into both the MLLM context and the Python runtime, enabling the agent to reference and manipulate the image during reasoning.

For video tasks, prior work typically relies on uniform frame sampling to construct the visual input. In contrast, `PyVision`-Video employs an on-demand context construction: The full video is loaded only into the Python runtime, and the agent is instructed via the system prompt to selectively sample and plot frames using Python code. This enables agentic frame fetching, where the agent dynamically chooses which frames to visualize based on the query or heuristic strategies. For example, for the query "`What is the actor doing in the last half of the video?`," the agent samples frames only from the latter portion of the video. This approach yields improved performance while substantially reducing visual token usage (Fig. 2).

### 3.2. Accumulative Tool Reward

Prior work observes that during RL training, the average number of tool calls tends to decrease steadily, often leading to a form of mode collapse where the model learns to invoke few or no tools (Hong et al., 2025; Zhang et al., 2025b). To enable stable RL training over hundreds or thousands of steps with sustained gains, and to prevent collapse in multi-turn tool usage, we introduce an RL objective with an accumulative tool reward. In addition to improving training stability, this reward explicitly incentivizes multi-turn tool usage, as demonstrated in Fig. 7.

Concretely, each rollout is evaluated using a combination of answer accuracy and tool usage. After a rollout is completed, we verify the correctness of the final answer, yielding an accuracy reward $R_{acc} \in \{0, 1\}$. In addition, we compute an accumulative tool reward proportional to the number of tool

*Table 1.* **Performance of PyVision-Image across diverse benchmarks.** We compare PyVision-Image with prior methods using either static toolsets or dynamic tooling, all based on Qwen2.5-VL-7B, across three task categories: visual search, multimodal reasoning, and agentic reasoning. PyVision-Image achieves state-of-the-art results in all three domains. For visual search, it improves over Qwen2.5-VL-7B by +10.2%, +6.5%, and +6.4% on V*, HRBench-4K, and HRBench-8K, respectively. For multimodal reasoning, it outperforms DeepEyes-v2 by +4.4%, +3.1%, and +9.6% on DynaMath, MathVerse, and WeMath. For agentic reasoning, it achieves a +7.3% gain on TIR-Bench over Qwen2.5-VL-7B. These results demonstrate the flexibility and broad effectiveness of dynamic tooling across diverse multimodal tasks. Results marked with † report avg@32.

| | **Visual Search** | | | **Multimodal Reasoning** | | | | **Agentic Reasoning** |
| | V* | HRBench-4K | HRBench-8K | DynaMath | MathVerse | MathVision | WeMath | TIR-Bench |
|---|---|---|---|---|---|---|---|---|
| Qwen2.5-VL-7B (Bai et al., 2025) | 78.5 | 71.6 | 67.9 | 53.3 | 45.6 | 25.6 | 34.6 | 16.0 |
| *Static Toolset* | | | | | | | | |
| Pixel-Reasoner (Su et al., 2025a) | 84.3 | 74.0 | 66.9 | - | - | - | - | - |
| Mini-o3 (Lai et al., 2025) | 88.2† | 77.5 | 73.3 | - | - | - | - | - |
| DeepEyes (Zheng et al., 2025c) | 85.6 | 75.1 | 72.6 | 55.0 | 47.3 | 26.6 | 38.9 | 17.3 |
| *Dynamic Tooling* | | | | | | | | |
| Thyme (Zhang et al., 2025b) | 82.2 | 77.0 | 72.0 | - | - | 27.6 | 39.3 | - |
| CodeV (Hou et al., 2025) | 84.8 | 76.1 | 71.3 | - | - | - | - | - |
| CodeDance (Song et al., 2025) | 84.8 | 75.2 | 72.3 | - | 46.8 | **29.6** | 39.6 | - |
| CodeVision (Guo et al., 2025b) | 83.7 | 75.6 | 72.2 | - | - | - | - | - |
| DeepEyes-v2 (Hong et al., 2025) | 81.8 | 77.9 | 73.8 | 57.2 | 52.7 | 28.9 | 38.1 | - |
| PyVision-Image | **88.7†** | **78.1** | **74.3** | **61.6** | **55.8** | 28.7 | **47.7** | **19.8** |

*Table 2.* **Performance comparison on VSI-Bench.** We compare PyVision-Video with Video-R1, a video understanding model using pure textual reasoning, and VITAL, an agentic video model with predefined video clipping tools. All methods are based on Qwen2.5-VL-7B and trained with RL. PyVision-Video achieves a 7.3% absolute improvement over the Qwen2.5-VL-7B baseline, demonstrating the effectiveness of dynamic tooling for spatial reasoning.

| | Avg. | Obj. Count | Abs. Dist. | Obj. Size | Room Size | Rel. Dist. | Rel. Dir. | Route Plan | Appr. Order |
|---|---|---|---|---|---|---|---|---|---|
| Qwen2.5-VL-7B (Bai et al., 2025) | 36.7 | 41.9 | 21.4 | 50.4 | 36.8 | 38.5 | 40.9 | 29.9 | 34.1 |
| Video-R1 (Feng et al., 2025) | 37.1 | - | - | - | - | - | - | - | - |
| VITAL (Zhang et al., 2025a) | 41.8 | - | - | - | - | - | - | - | - |
| PyVision-Video | **44.0** | 53.8 | 25.8 | 50.8 | 38.2 | 44.8 | 46.3 | 26.3 | 58.6 |

calls, given by $0.1 \cdot n_{tc}$, where $n_{tc}$ denotes the total number of tool calls during the rollout. This accumulative tool reward is added to the final reward only when the answer is correct, ensuring that tool usage is encouraged without rewarding unproductive or incorrect tool calls.

The final RL objective is as below:

$$R = R_{acc} + \underbrace{0.1 \cdot n_{tc} \cdot \mathbf{1}_{\{R_{acc}=1\}}}_{\text{accumulative tool reward}} \quad (1)$$

### 3.3. Oversampling–Filtering–Ranking Rollouts

When extending vanilla GRPO from pure textual reasoning to agentic RL, rollout quality and distribution become a dominant factor for training stability and efficiency. In practice, we observe that a significant portion of generated rollouts either provides little learning signal or actively destabilizes training. For example, when a query is too difficult for the current policy, all rollouts within a group may receive zero reward, resulting in zero advantages after group-level normalization and contributing no gradient to learning. Similarly, under our reward design, groups where all rollouts

are correct but have identical tool-call counts also collapse to zero advantage, effectively wasting training compute.

A second challenge arises from the inherent uncertainty of agent–environment interaction. During rollout generation, the agent may produce invalid or non-executable Python code due to timeouts, runtime failures, or invalid multimodal outputs, *e.g.*, exceeding image limits or failing to render any image. Such broken trajectories can interrupt or crash the RL training if not handled properly, observed also in prior agentic RL works (Xue et al., 2025; Luo et al., 2025). To ensure stable training, it is therefore necessary to detect and exclude malformed rollouts before policy optimization.

Finally, even among valid and correct rollouts, reward shaping can introduce subtle optimization issues. In particular, when multiple correct trajectories exist within a group but differ in tool-call counts, group-level normalization may assign negative advantages to correct but more concise solutions, suppressing useful behaviors during training.

To address these challenges, we adopt an oversampling, filtering, and ranking framework for rollout generation. Specif-

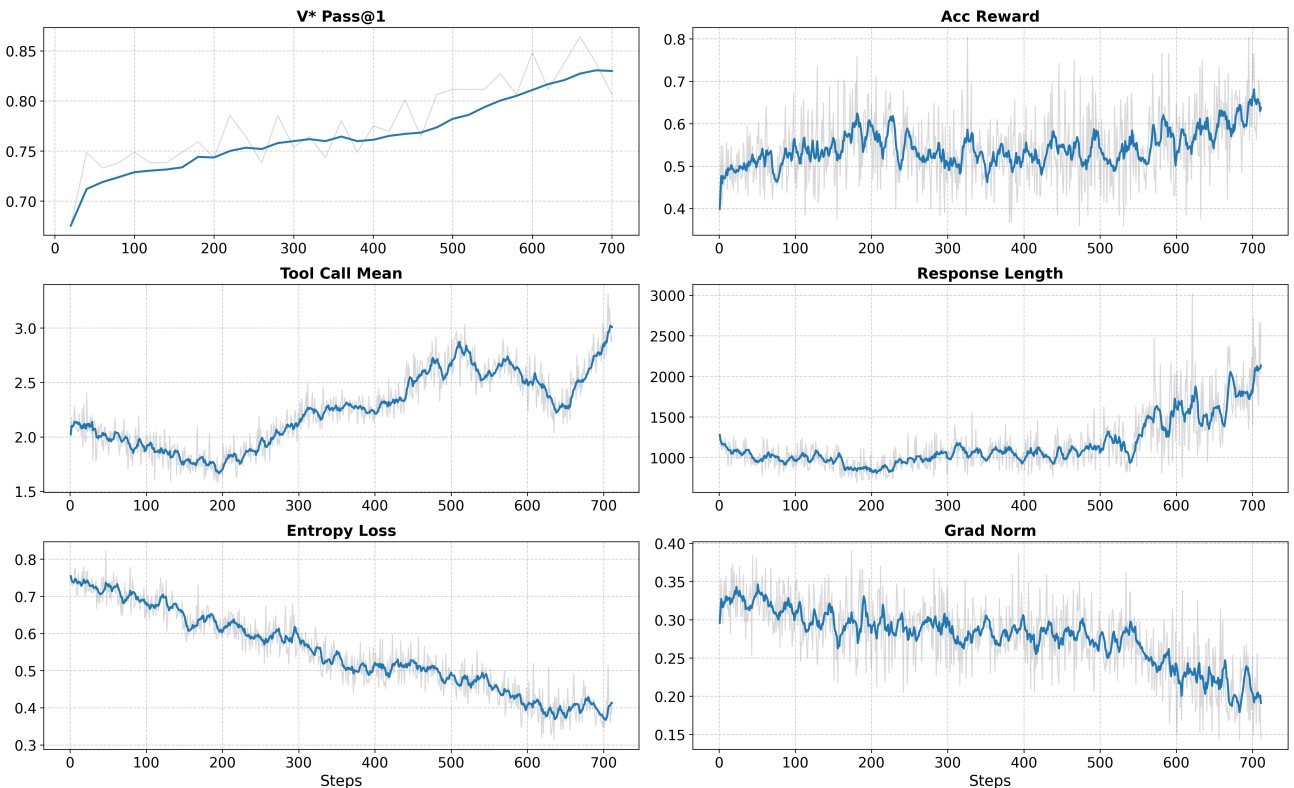

*Figure 3.* **Training dynamics of RL for PyVision-Image.** Our training algorithm yields stable optimization and steadily improving performance. Entropy loss and gradient norm decrease smoothly over training, indicating stable RL dynamics. Meanwhile, validation performance on V*, accuracy reward, response length, and the mean number of tool calls consistently increase, showing that the model learns sustained, long-horizon tool-using behavior.

ically, we first oversample rollouts, then apply online filtering to remove groups with zero reward variance and rollouts with broken agent–environment interaction. Among the remaining candidates, we rank rollout groups by group-level reward standard deviation, which serves as a proxy for sample difficulty (Jiang et al., 2024; Zhu et al., 2025), and retain the top-ranked groups for training. This strategy prioritizes moderately difficult rollouts that provide informative learning signals, while also substantially reducing the prevalence of correct samples with negative advantages, resulting in more stable and efficient agentic RL (Sec. 4.3). We refer to this strategy as Standard Deviation Sorting.

### 3.4. Optimization and Data Collection

**Removing Standard Deviation Normalization in GRPO.**
We adopt GRPO (Shao et al., 2024) as the base algorithm for RL training. Let $\pi_\theta$ denote the policy model, and let $x$ be sampled from the training dataset $\mathcal{D}$. For each input $x$, we generate $G$ rollouts $y_{i_{i=1}^G}$ and compute rewards at the rollout level. Different from the original GRPO, however, we remove the standard deviation normalization term in the intra-group advantage computation, following recent works on improving training stability and performance in

LLM RL (Luo et al., 2025; Liu et al., 2025a;c; Zheng et al., 2025a). The advantage for each token is computed as:

$$\widehat{A}_{i,t} = R(x, y_i) - \text{mean}\left(\{R(x, y_i)\}_{i=1}^G\right). \quad (2)$$

where $R(x, y_i)$ denotes the rollout-level reward. We empirically verify the effectiveness of removing standard deviation normalization in Sec. 4.2.

**SFT Data Collection and Training.** We first obtain SFT models as a cold start to endow the base models with basic multi-turn tool-using capabilities. Specifically, we train PyVision-Image-SFT using synthetic data generated with GPT-4.1 (Zhao et al., 2025a). To ensure broad generalization of multi-turn tool use across domains, the SFT data spans multimodal reasoning (MMK12 (Meng et al., 2025a)), medical reasoning (GMAI-Reasoning (Su et al., 2025b)), chart understanding (ChartQA (Masry et al., 2022), InfoVQA (Mathew et al., 2022)), and general visual question answering (MMPR (Wang et al., 2024b)). We filter out samples with incorrect answers or fewer than two tool-use turns, resulting in 7K high-quality SFT examples that emphasize sustained interaction.

For PyVision-Video-SFT, on-demand context construction represents a novel capability absent from the base

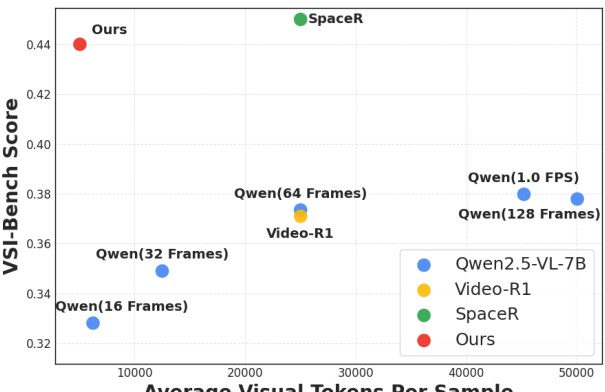

*Figure 4.* **Efficiency performance trade-off on VSI-Bench.** Thanks to on-demand context construction, PyVision-Video selectively samples task-relevant frames during reasoning, achieving higher accuracy with substantially fewer visual tokens compared to frame-sampling baselines such as Qwen2.5-VL series.

model. We therefore curate a SFT dataset consisting of 44K samples, covering spatial reasoning (Ouyang et al., 2025) and long-video reasoning (Chen et al., 2025b; 2024), using the same synthesis and filtering pipeline as for images. Both SFT models are trained using LLaMA-Factory (Zheng et al., 2024) on a single node for one epoch.

**RL Data Collection and Training.** After initializing the models with SFT, we further apply reinforcement learning to specialize agentic behavior. For PyVision-Image, RL training focuses on visual search and multimodal reasoning tasks. We collect 44K visual search samples from Deep-Eyes (Zheng et al., 2025c) and Mini-o3 (Lai et al., 2025), and multimodal reasoning data from V-Thinker (Qiao et al., 2025b) and WeMath (Qiao et al., 2025c). For PyVision-Video, we focus on spatial reasoning and collect 15K samples from SpaceR (Ouyang et al., 2025). Detailed data composition statistics are provided in Appendix Sec. B.2.

PyVision-Image is built on Qwen2.5-VL-7B, which requires resizing extremely small or large images prior to input. Following Mini-o3 (Lai et al., 2025), we control image resizing using two thresholds, with `min_pixels` set to 3,136 and `max_pixels` set to 2,000,000, enabling efficient handling of high-resolution images.

Both PyVision-Image and PyVision-Video are trained for 700 RL steps using the same hyperparameters: oversampling batch size 32, training batch size 16, group size 8, and learning rate $1 \times 10^{-6}$ on 8 H100 GPUs.

## 4. Experiments

**Evaluation Setup.** During evaluation, PyVision-Image uses a temperature of 0.01 for V* and 0.5 with top-k 20 for the other benchmarks, whereas PyVision-Video uses

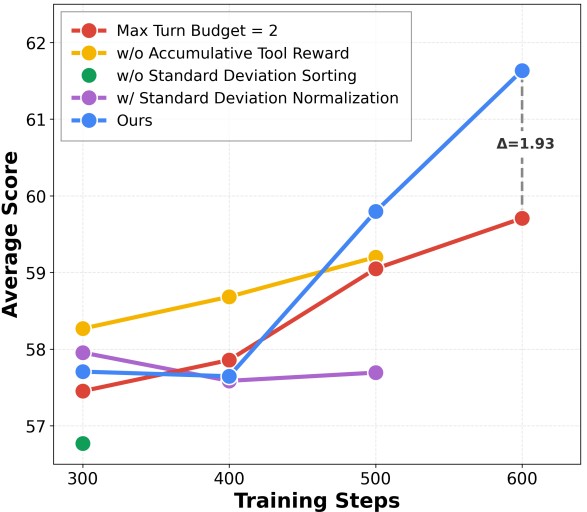

*Figure 5.* **Ablation of training components.** We report the average performance over seven benchmarks (V* avg@32, HRBench-4K, HRBench-8K, MathVision, MathVerse, WeMath, and DynaMath) under different training configurations, each ablating one component of our method. The *Ours* setting uses a max turn budget of 4, includes the accumulative tool reward, applies standard deviatio sorting for rollout groups, and removes standard deviation normalization term in advantage estimation. All other settings modify exactly one component relative to *Ours*. Overall, we observe that (1) applying standard deviation sorting or removing standard deviation normalization consistently improves performance, and (2) incorporating the accumulative tool reward or increasing the max turn budget leads to larger performance gains in later training stages. For example, at step 600, a max turn budget of 4 outperforms a budget of 2 by 1.93%.

a temperature of 0.01. Given the long-horizon reasoning capabilities induced by RL tuning, we set the maximum turn budget to 30 and the maximum context length to 32K tokens. We evaluate our models on the following benchmarks:

*Visual Search.* To assess the model's agentic visual perception capabilities, we evaluate our model on V* (Wu & Xie, 2024), HRBench-4K (Wang et al., 2025b), and HRBench-8K (Wang et al., 2025b). Since V contains only 191 samples, we report results using the avg@32 metric.

*Multimodal Reasoning.* We evaluate PyVision-Image on multimodal math benchmarks, including MathVerse (Zhang et al., 2024), MathVision (Wang et al., 2024a), We-Math (Qiao et al., 2025a), and DynaMath (Zou et al., 2024).

*Agentic Reasoning.* TIR-Bench (Li et al., 2025b) consists of tasks that *require* multi-turn tool usage. We evaluate PyVision-Image on this benchmark to assess its agentic reasoning and the effectiveness of dynamic tooling.

*Spatial Reasoning.* We benchmark PyVision-Video on VSI-Bench (Yang et al., 2024) for its spatial reasoning capability given a video of an enviroment.

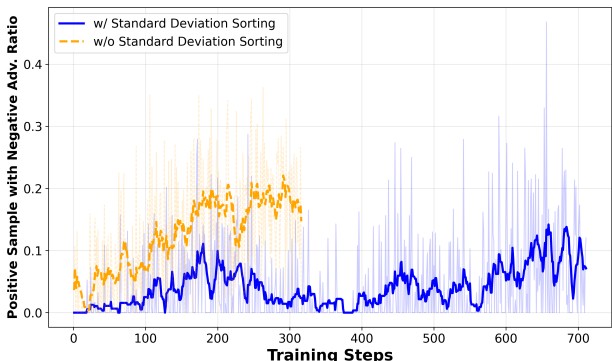

*Figure 6.* **Ratio of positive samples with negative advantage.** Positive samples with negative advantage are correct trajectories that receive negative advantages due to relatively fewer tool calls within a group. We compare the proportion of such samples in each training batch with and without standard-deviation-based rollout sorting. Applying standard deviation sorting significantly reduces this ratio throughout training.

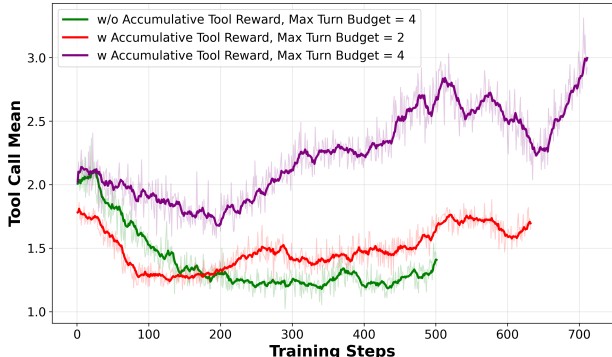

*Figure 7.* **Mean number of tool calls during RL training.** We ablate the accumulative tool reward and the max turn budget. Without the accumulative tool reward, the average number of tool calls rapidly decreases and stabilizes at a low value. In contrast, incorporating the accumulative tool reward encourages sustained tool usage, with higher max turn budgets leading to a larger and faster increase in tool calls.

### 4.1. Main Results

**Strong Performance on Image Benchmarks.** Tab. 1 summarizes the performance of `PyVision`-Image on visual search, multimodal reasoning, and agentic reasoning benchmarks. The compared methods fall into two categories: (1) models trained with a predefined static toolset (e.g., crop and zoom-in), including Pixel-Reasoner (Su et al., 2025a), Mini-o3 (Lai et al., 2025), and DeepEyes (Zheng et al., 2025c; Hong et al., 2025), and (2) models that use a Python interpreter as the primitive tool, including Thyme (Zhang et al., 2025b), CodeV (Hou et al., 2025), CodeDance (Song et al., 2025), CodeVision (Guo et al., 2025b), and DeepEyes-v2 (Hong et al., 2025). Our method adopts the latter.

`PyVision`-Image consistently achieves strong performance across all evaluated tasks. On visual search benchmarks, it outperforms all competing methods, yielding absolute improvements of +10.2%, +6.5%, and +6.4% on V*, HRBench-4K, and HRBench-8K, respectively, compared to the base model Qwen2.5-VL-7B. These results indicate that `PyVision`-Image substantially enhances fine-grained visual localization and agentic perception capabilities.

On multimodal reasoning benchmarks, `PyVision`-Image establishes new state-of-the-art results on DynaMath, Math-Verse, and WeMath, surpassing the previous best model, DeepEyes-v2, by +4.4%, +3.1%, and +9.6%, respectively. This demonstrates that the gains from agentic RL extend beyond perception-oriented tasks and translate effectively to complex multimodal mathematical reasoning.

Finally, on agentic reasoning tasks requiring multi-turn tool usage, `PyVision`-Image improves performance by +3.8% over the base model, highlighting the effectiveness of dynamic tool invocation for long-horizon reasoning.

**Token Efficiency on Video Benchmarks.** Fig. 2 contrasts the conventional video processing strategy adopted by most MLLMs, where they uniformly sample frames from the input video, with the on-demand frame retrieval used in `PyVision`-Video. Rather than committing to a fixed frame sampling rate, `PyVision`-Video dynamically queries the video through Python code, extracts informative key frames from the full frame sequence based on model's reasoning, and selectively includes them in the MLLM context. This on-demand context construction eliminates redundant visual tokens while preserving task-relevant information.

Quantitatively, Fig. 4 compares the average of visual tokens consumed per sample on VSI-Bench across `PyVision`-Video, Qwen2.5-VL-7B, Video-R1 (Feng et al., 2025), and SpaceR (Ouyang et al., 2025). `PyVision`-Video uses approximately 5K visual tokens per sample on average, achieving a performance of 44.0%. In contrast, Qwen2.5-VL-7B attains its best performance (38.0%) when sampling at 1.0 FPS, at the cost of approximately 45K visual tokens per sample. Video-R1 and SpaceR reduce token usage to around 25K per sample, with SpaceR achieving comparable performance (45.6%) to `PyVision`-Video. Overall, `PyVision`-Video achieves the most favorable trade-off between visual token efficiency and reasoning performance on VSI-Bench, demonstrating that agentic, on-demand frame selection can substantially reduce context length without sacrificing accuracy. Overall, `PyVision`-Video achieves the most favorable trade-off between visual token efficiency and reasoning performance, demonstrating that agentic, on-demand frame selection can substantially reduce context length without sacrificing accuracy.

Tab. 2 shows the per-category results on VSI-Bench (Yang et al., 2024). `PyVision`-Video outperforms Video-R1 and

VITAL, and makes a performance improvement of +7.3% compared with Qwen2.5-VL-7B. We further illustrate qualitative examples in Figs. 19 and 20, which visualize how `PyVision`-Video identifies and incorporates only the most informative frames for spatial reasoning.

## 4.2. Ablation Study

To evaluate the contribution of each component in our method, we conduct a comprehensive ablation study, examining the effects of the maximum turn budget, accumulative tool reward, standard deviation sorting and removing standard deviation normalization during RL training. Our final training algorithm is used as the baseline, and we ablate by *removing* one component at a time. The overall ablation results are summarized in Fig. 5.

**Max Turn Budget.** We first examine the impact of the maximum turn budget on model performance. In our baseline setting, the maximum turn budget is set to 4, and we compare it against a reduced setting of 2 turns. During the early stages of RL training (e.g., at 300 or 400 steps), increasing the turn budget does not lead to immediate performance gains. However, as training progresses, the benefit of a larger turn budget becomes apparent: At 600 training steps, the model trained with a maximum turn budget of 4 significantly outperforms the one trained with a budget of 2. This suggests that a larger turn budget increases the performance upper bound of the model, with its advantages emerging in later stages of RL optimization.

**Accumulative Tool Reward.** Next, we study the effect of the accumulative tool reward. In the baseline, we apply an accumulative tool reward with a coefficient of 0.1 during RL training ( Eq. (1)). To ablate its effect, we rerun training with the coefficient set to 0. Removing the accumulative tool reward leads to a noticeable reduction in tool usage during training, as illustrated in Fig. 7. In Fig. 5, the model without the accumulative tool reward achieves slightly better performance in the early stage of RL training. However, as training continues to beyond 500 steps, its performance falls behind the baseline. This indicates that while the accumulative tool reward may slow early optimization, it plays a crucial role in enabling stronger long-horizon reasoning and improved final performance.

**Standard Deviation Sorting and Normalization.** Finally, we analyze standard deviation sorting and normalization. Removing standard deviation sorting during RL training degrades performance in the early stages, as shown in Fig. 5, indicating its importance for stabilizing optimization when rewards are noisy. Meanwhile, retaining the common standard deviation normalization in the advantage computation leads to persistent performance fluctuations as training progresses, suggesting that it introduces excessive variance into the learning dynamics and hampers convergence.

## 4.3. Analysis

**RL Training Dynamics.** We visualize the RL training dynamics of `PyVision`-Image in Fig. 3. Under our training algorithm, the optimization process remains stable throughout training: entropy loss and gradient norm decrease steadily, while the mean number of tool calls, accuracy reward, and response length consistently increase. The growth in tool usage and response length indicates that RL successfully incentivizes sustained multi-turn interaction within each episode. In addition, the validation performance on V∗ improves monotonically during training, demonstrating effective generalization.

**How Does Standard Deviation Sorting Work?** Our ablation shows that removing Standard Deviation Sorting leads to a significant performance drop (Fig. 5), indicating that this component plays an important role in training. We provide two complementary explanations for its effectiveness.

First, from a curriculum learning perspective, group-level standard deviation serves as a proxy for sample difficulty. Groups with higher reward variance typically contain both correct and incorrect rollouts, corresponding to cases that are neither trivially easy nor excessively difficult for the current policy. In contrast, groups where all rollouts are correct or all are incorrect exhibit low variance and provide limited learning signal. By prioritizing groups with higher standard deviation, Standard Deviation Sorting encourages the policy to learn from moderately difficult samples that are most informative at the current training stage, consistent with curriculum learning principles (Jiang et al., 2024).

Second, Standard Deviation Sorting mitigates the prevalence of *positive samples with negative advantages*. These samples correspond to correct rollouts that receive negative advantages due to relatively fewer tool calls within their group. Although correct, such samples are suppressed during policy updates, leading to compression of desirable behaviors. As shown in Fig. 6, applying Standard Deviation Sorting significantly reduces the proportion of these samples throughout training. This indicates that the method improves optimization not only by selecting informative samples, but also by suppressing adverse gradient signals caused by group-level normalization effects.

## 5. Conclusion

We present `PyVision`-RL, a unified agentic multimodal framework for image and video understanding that adopt Python for dynamic tooling. To stabilize tool-use RL, we introduce an oversampling–filtering–ranking framework for rollout generation, and show increasing the max turn budget leads to a higher performance ceiling. Empirically, `PyVision`-Image achieves strong performance across benchmarks, outperforming prior agentic MLLMs.

PyVision-Video shows effective spatial reasoning while substantially reducing visual token usage, achieving a favorable accuracy–efficiency trade-off on VSI-Bench. Together, these results highlight the effectiveness of dynamic tooling and sustained interaction for multimodal agentic reasoning.

## Impact Statement

In this paper, we present PyVision-Image and PyVision-Video, two agentic vision models capable of doing image and video understanding tasks. These two models enhance the multi-modal agents development. But, since these models use Python as the primitive tool, it may access the host file system and makes damage. Thus, the deployments of PyVision-Image and PyVision-Video needs careful consideration of these impacts.

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

## Appendix Contents

## A. System Prompts

### A.1. System Prompt of **PyVision**-Image

We illustrate the system prompt of PyVision-Image in Fig. 8.

### A.2. System Prompt of **PyVision**-Video

We illustrate the system prompt of PyVision-Video in Fig. 9.

## B. More Details of Training Pipeline and Training Data

### B.1. Illustration of Oversampling-Filtering-Ranking Framework for Rollout Generation

The detail of oversampling-filtering-ranking rollout generation and training pipeline is shown in Fig. 10 and Algorithm. 1.

---

**Algorithm 1** Oversampling-Filtering-Ranking Framework for Rollout Generation

---

**Input:** Prompt pool $\mathcal{P}$, batch size $B$, group size $G$, oversampling ratio $\alpha > 1$, policy $\pi_\theta$, reward model $\mathcal{R}$
**Output:** Selected rollout batch $\mathcal{D}_{\text{train}}$ for policy update
Sample $\alpha B$ prompts $\{x_j\}_{j=1}^{\alpha B}$ from $\mathcal{P}$ {Oversampling stage}
**for** $j = 1$ **to** $\alpha B$ **do**
    Generate $G$ rollouts $\{o_{j,i}\}_{i=1}^{G} \sim \pi_\theta(\cdot|x_j)$ via Rollout Worker
    Execute code blocks in environment and receive observations
    **if** any rollout encounters timeout, runtime death, or execution error **then**
        Mark as broken trajectory
    **end if**
    Compute rewards $r_{j,i} = \mathcal{R}(x_j, o_{j,i})$ for each rollout
    Compute group statistics: $\mu_{j,i} = \frac{1}{G}\sum_{i=1}^{G} r_{j,i}$, $\sigma_{j,i} = \sqrt{\frac{1}{G}\sum_{i=1}^{G}(r_{j,i} - \mu_{j,i})^2}$
**end for**
Initialize filtered set $\mathcal{F} = \emptyset$
**for** $j = 1$ **to** $\alpha B$ **do**
    **for** $i = 1$ **to** $G$ **do**
        **if** all rollouts $o_{j,i}$ is broken **then**
            **continue** {Filter $o_{j,i}$}
        **end if**
        **if** $\sigma_{j,i} = 0$ **then**
            **continue** {Filter $o_{j,i}$}
        **end if**
        Add rollout $o_{j,i}$ to $\mathcal{F}$
    **end for**
**end for**
Sort $\mathcal{F}$ by group-level std $\sigma_{j,i}$ in descending order {Ranking via difficulty}
Select top $B * G$ samples from sorted $\mathcal{F}$ as $\mathcal{D}_{\text{train}}$ {Select moderately difficult samples}

---

### B.2. Training Data Distribution

We illustrate the SFT and RL data of PyVision-Image and PyVision-Video in Fig. 11 and Fig. 12.

## C. More Evaluation Results

### C.1. Ablation Results Plot on Different Benchmarks

We plot the results across different benchmarks under different training settings, in Fig. 13

*Table 3.* **The details of the ablation of training components.** We ablate four conponents used in our training pipeline, i.e., accumulative tool reward (ATR), standard deviation ranking (SRK), removing standard deviation normalization in advantage estimation (RSN), maximum turn budget (MTB). First, for maximum turn budget, a larger one makes a better performance at later training stage, i.e., maximum turn budget of 4 outperforms that of 2 by +1.77% on V* and +4.65% on MathVerse at training step 600. For accumulative tool reward, adding it to the RL objective makes performance gain by +1.91% on V*, +1.63% on HRBench-4K, +1.00% on HRBench-8K, at training step 500. For stantard deviation sorting, it improves the performance by +2.26% on HRBench-4K, +1.90% on WeMath, at training step 300. For standard deviation normalization term, removing them improve the performance by +4.94% on V*, +2.75% on HRBench-4K, +3.62% on WeMath, at training step 500.

| | | | | | | Visual Search | | Multi-modal Reasoning | | | |
| --- | --- | --- | --- | --- | --- | --- | --- | --- | --- | --- | --- |
| | | | | | V* | HRBench-4K | HRBench-8K | MathVision | MathVerse | WeMath | DynaMath |
| **PyVision-Image-SFT** | | | | | 75.98 | 73.25 | 66.75 | 25.07 | 47.23 | 31.90 | 58.64 |
| Steps | ATR | SRK | RSN | MTB | | | | | | | |
| 300 | ✓ | ✓ | ✗ | 4 | 82.07 | 75.62 | 69.87 | 27.96 | 49.44 | 40.67 | 60.05 |
| | ✓ | ✗ | ✓ | 4 | 81.61 | 73.62 | 67.75 | 26.91 | 49.57 | 37.43 | 60.50 |
| | ✗ | ✓ | ✓ | 4 | 80.51 | 74.75 | 71.25 | 27.86 | 51.78 | 41.90 | 59.82 |
| | ✓ | ✓ | ✓ | 2 | 81.50 | 73.12 | 71.25 | 25.03 | 50.48 | 41.14 | 59.64 |
| | ✓ | ✓ | ✓ | 4 | 81.95 | 75.88 | 68.50 | 27.20 | 51.50 | 39.33 | 59.58 |
| 400 | ✓ | ✓ | ✗ | 4 | 80.96 | 73.88 | 68.50 | 25.86 | 50.38 | 43.24 | 60.28 |
| | ✗ | ✓ | ✓ | 4 | 81.81 | 76.00 | 69.25 | 28.22 | 52.82 | 42.76 | 59.92 |
| | ✓ | ✓ | ✓ | 2 | 83.12 | 74.12 | 70.13 | 27.07 | 50.89 | 40.67 | 59.00 |
| | ✓ | ✓ | ✓ | 4 | 82.05 | 74.50 | 68.75 | 27.02 | 50.13 | 40.57 | 60.50 |
| 500 | ✓ | ✓ | ✗ | 4 | 81.41 | 74.50 | 69.87 | 27.47 | 52.20 | 38.48 | 59.92 |
| | ✗ | ✓ | ✓ | 4 | 84.44 | 75.62 | 70.63 | 28.22 | 52.87 | 41.62 | 61.00 |
| | ✓ | ✓ | ✓ | 2 | 83.92 | 73.38 | 70.13 | 26.97 | 51.80 | 43.33 | 63.81 |
| | ✓ | ✓ | ✓ | 4 | **86.35** | 77.25 | 71.63 | 27.80 | 53.20 | 42.10 | 60.24 |
| 600 | ✓ | ✓ | ✓ | 2 | 84.47 | 76.38 | 71.37 | **28.67** | 52.66 | 44.38 | 60.02 |
| | ✓ | ✓ | ✓ | 4 | 86.24 | **77.72** | **72.22** | 28.66 | **57.31** | **47.71** | **61.58** |

## C.2. Ablation Results Detail

Besides the plot, we list the exact ablation result number in Tab. 3.

# D. More Analysis

## D.1. Training Dynamics of PyVision-Video

We visualize the training dynamics of PyVision-Video in Fig. 14.

## D.2. Why Tool Call Count Increasing During RL?

In Fig. 15, we visualize the average number of tool using and the ratio of positive samples with negative advantage during RL. We find a negative correlation between these two metrics. Thus, based on this observation, we think the tool call mean increasing comes from the negative singnals of the correct samples with relatively fewer tool calls.

## D.3. Tool Category Distribution

Based on the tooling taxonamy presented in PyVision (Zhao et al., 2025a), we illustrated the tooling categories distribution of PyVision-Image on different benchmarks in Fig. 21.[1] Also, we present the tooling categories distribution in Fig. 23.

## D.4. Tool Call Numbers Distribution

We present tool call numbers of PyVision-Image in Fig. 22 and PyVision-Video in Fig. 24.

---

[1]Since there are many operations, which are just plot the original images, we remove these part from Fig. 21. For the full tooling distribution, see Fig. 25.

**D.5. Case Study**

D.5.1. CASE STUDY OF PYVISION-IMAGE

We visualize two examples of the reasoning process of PyVision-Image on TIR-Bench in Fig. 17 and Fig. 18.

D.5.2. CASE STUDY OF PYVISION-VIDEO

We visualize two examples of the reasoning process of PyVision-Video on VSI-Bench in Fig. 19 and Fig. 20.

---

**System Prompt Template of `PyVision`-Image**

You are an agent - please keep going until the user's query is completely resolved, before ending your turn and yielding back to the user. Only terminate your turn when you are sure that the problem is solved.

Solve the following problem step by step. You now have the ability to selectively write executable Python code to enhance your reasoning process. The Python code will be executed by an external sandbox.

You MUST plan extensively before each function call, and reflect extensively on the outcomes of the previous function calls. DO NOT do this entire process by making function calls only, as this can impair your ability to solve the problem and think insightfully.

For all the provided images, in order, the i-th image has already been read into the global variable `"image_clue_i"` using the `"PIL.Image.open()"` function. When writing Python code, you can directly use these variables without needing to read them again.

Since you are dealing with the vision-related question answering task, you MUST use the python tool (e.g., matplotlib library) to analyze or transform images whenever it could improve your understanding or aid your reasoning. This includes but is not limited to zooming in, rotating, adjusting contrast, computing statistics, or isolating features.

Note that when you use matplotlib to visualize data or further process images, you need to use `"plt.show()"` to display these images; there is no need to save them. Do not use image processing libraries like cv2 or PIL. If you want to check the value of a variable, you MUST use `"print()"` to check it.

The output (wrapped in `"<interpreter>output_str</interpreter>"`) can be returned to aid your reasoning and help you arrive at the final answer. The Python code should be complete scripts, including necessary imports.
Each code snippet is wrapped with:

python code snippet


The last part of your response should be in the following format:
<answer>
\boxed{"The final answer goes here."}
</answer>

*image resolution:*
Image Width: {width}; Image Height: {height}
*user question:*
Answer the following Problem with an image provided and put the answer in the format of \boxed{answer}
{"query"}

Remember to place the final answer in the last part using the format:
<answer>
\boxed{"The final answer goes here."}
</answer>

*Figure 8*

---

**System Prompt Template of `PyVision`-Video**

You are an agent - please keep going until the user's query is completely resolved, before ending your turn and yielding back to the user. Only terminate your turn when you are sure that the problem is solved.

Solve the following problem step by step. You now have the ability to selectively write executable Python code to enhance your reasoning process. The Python code will be executed by an external sandbox.

You MUST plan extensively before each function call, and reflect extensively on the outcomes of the previous function calls. DO NOT do this entire process by making function calls only, as this can impair your ability to solve the problem and think insightfully.

For all the provided videos, in order, the j-th video has already been read into the global variable `"video_clue_j"` using the `"VideoReader()"` function. When writing Python code, you can directly use these variables without needing to read them again.

Since you are dealing with the vision-related question answering task, you MUST use the Python tool (e.g., matplotlib library) to analyze or transform images whenever it could improve your understanding or aid your reasoning. This includes but is not limited to zooming in, rotating, adjusting contrast, computing statistics, or isolating features. For the videos, you can also use the Python tool (e.g., decord library) to sample frames from the video, helping your reasoning.

Note:
1. When you use matplotlib to visualize data or further process images, you need to use `"plt.show()"` to display these images; there is no need to save them.
2. Do not use image processing libraries like cv2 or PIL.
3. Remember you CAN NOT see the video directly. Thus, if you need to reason based on the video, you MUST sample frames and use `"plt.show()"` to display these frames, helping your reasoning.
4. If you want to check the value of a variable, you MUST use `"print()"` to check it.
5. If you think the init provided frames are not enough to solve the question. just sample more fames from the `"video_clue_0"` using Python code.

The output (wrapped in `"<interpreter>output_str</interpreter>"`) can be returned to aid your reasoning and help you arrive at the final answer. The Python code should be complete scripts, including necessary imports.
Each code snippet is wrapped with:

python code snippet


*Video Information:*
{video_info}

*User Question:*
Answer the following Problem with an image provided and put the answer in the format of \boxed{answer}
{query}

Remember to place the final answer in the last part using the format:
<answer>
\boxed{'The final answer goes here.'}
</answer>

*Figure 9*

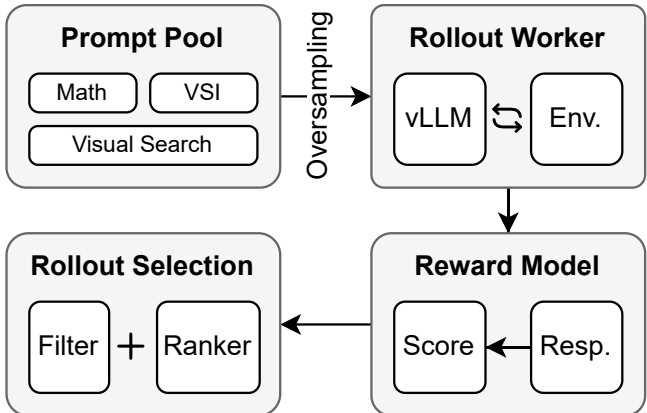

*Figure 10.* **Overview of the Oversampling-Filtering-Ranking Framework for Rollout Generation.** First, we oversample $\alpha * B$ prompts from the prompt pool, where $B$ is the batchsize and $\alpha$ is the oversampling parameter. Then, each prompt is sent to rollout worker to generate $G$ rollouts, where $G$ is the group size in the GRPO-like RL algorithms. In the generated rollouts, some of them are broken. For these $\alpha * B * G$ rollouts, we give their reward with reward model and calculate each one's group-level stantard deviation. Based on if it is broken and its group-level standard deviation, we filter and sort these rollouts, and keep top-$B * G$ rollouts as the training samples.

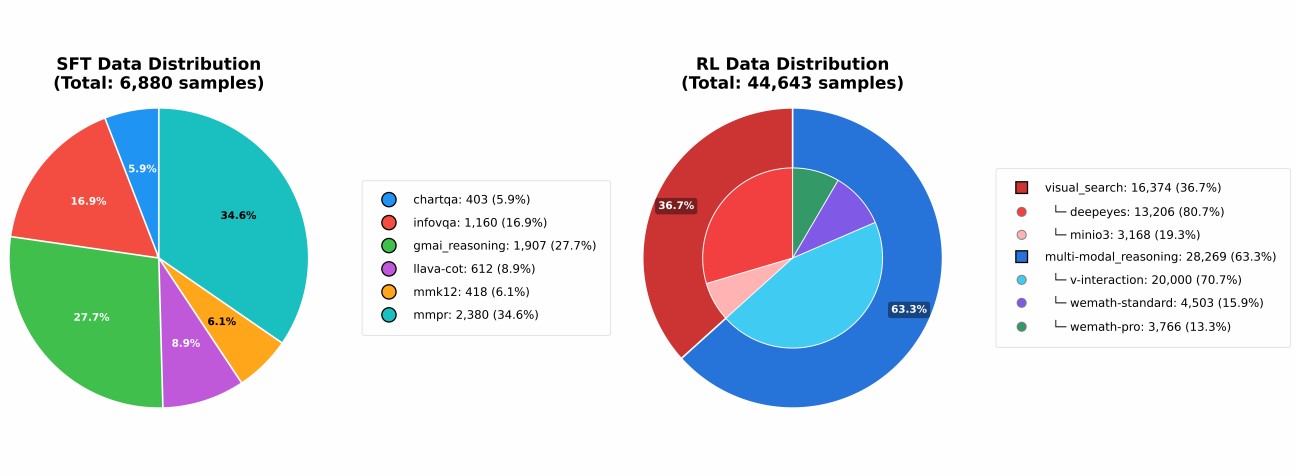

*Figure 11.* **Left**: we illustrate the distribution of SFT data of `PyVision`-Image, containing chart understanding data, from ChartQA, infografic understanding data, from InfoVQA, medical understanding data, from GMAI-Reasoning, math data, from MMK-12, and general VQA data, from LLaVA-CoT and MMPR. **Right**: we illustrate the RL data distribution of `PyVision`-Image, containing visual search data, from DeepEyes and Mini-o3, and multi-modal reasoning data, from V-Thinker and WeMath-v2.

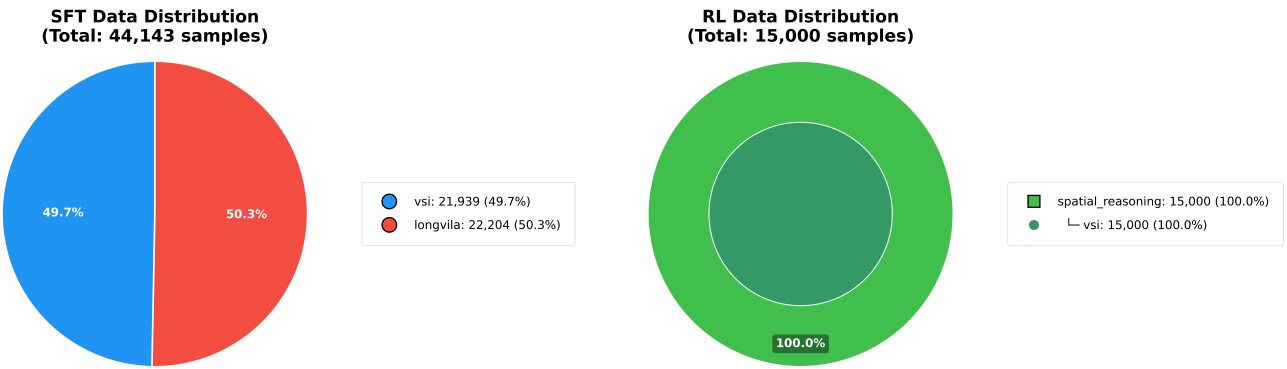

*Figure 12.* **Left**: we illustrate the distribution of SFT data of `PyVision`-Video, containing visual spatial reasoning data, from SpaceR, and long video understanding data, from LongVILA. **Right**: the RL data used in `PyVision`-Video training is all visual spatial reasoning data, from SpaceR.

![Figure 13 performance comparison charts]

*Figure 13.* Performance Comparison of Different RL Training Settings.

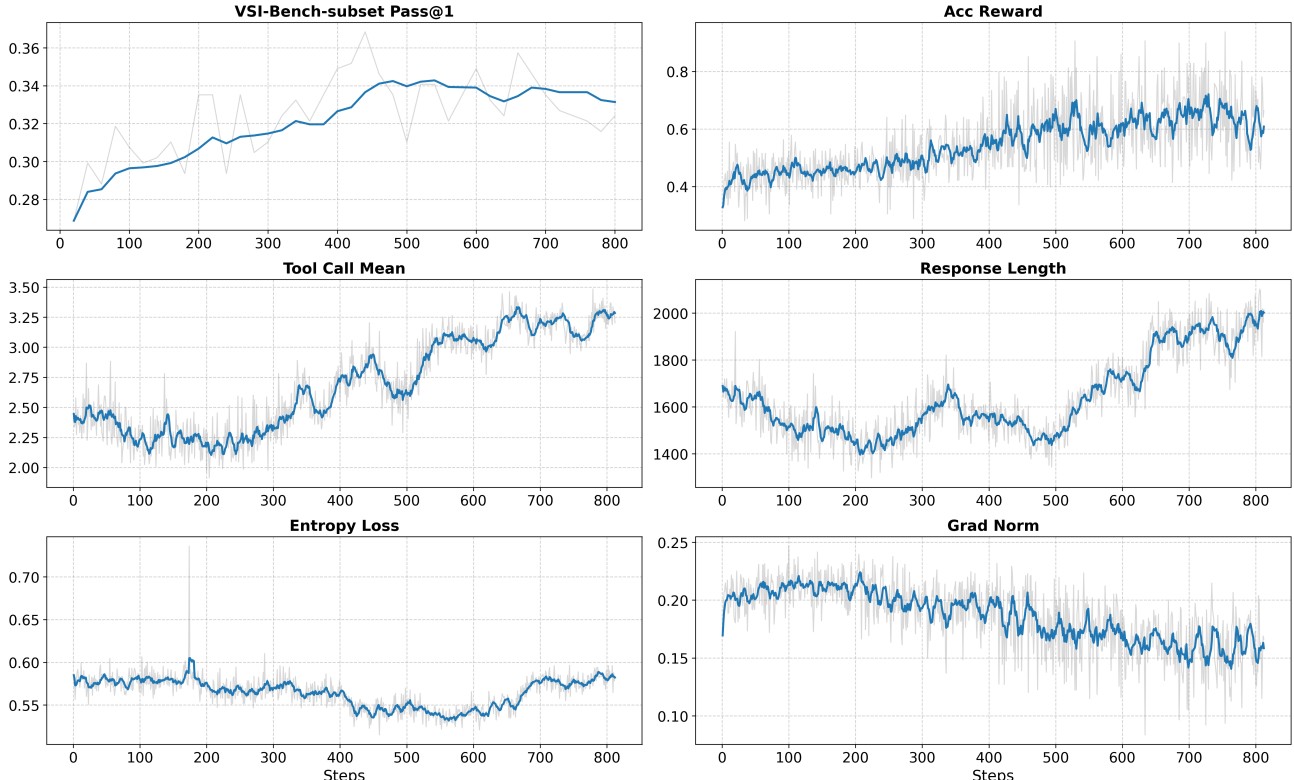

*Figure 14.* **Training dynamics of PyVision-Video's RL process.** Our algorithm makes a stable training and a continuous performance increasing. Entropy loss keeps in a moderate level and grad norm decrease steadily, indicating stable RL optimization. Vilidation score on VSI-Bench-subset, accuracy reward, response length and the average tool call numbers increase steadily during RL, showing that the model learns sustained, long-horizon tool-using behavior. To make validation efficient during training, we sample 400 samples randomly from VSI-Bench as the validation dataset, named as VSI-Bench-subset.

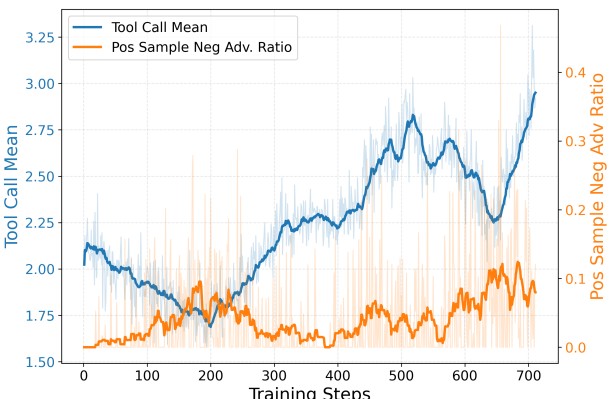

*Figure 15.* **The average number of tool calling and the ratio of positive samples with negative advantage.** We visualize the tool call mean curve and positive sample with negative advantage ratio curve of PyVision-Image. These two metrics are negatively correlated. Inspired by this observation, we hypothesize that the main reason of tool call mean increasing comes from the negative signals of the correct samples but using relatively fewer tools.

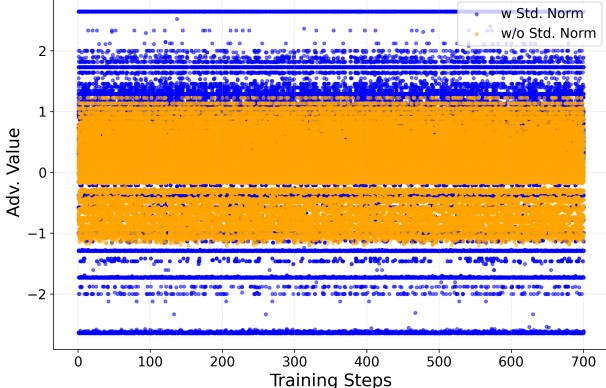

*Figure 16.* **Advantage distribution of w/ and w/o standard deviation normalization term in advantage estimation.** In our experiments, advantage estimated without standard deviation normalization term makes the performance improving more stably. We compare the advantage distribution calculated with and without this term – advantage without it presents lower variance, making RL training more stable.

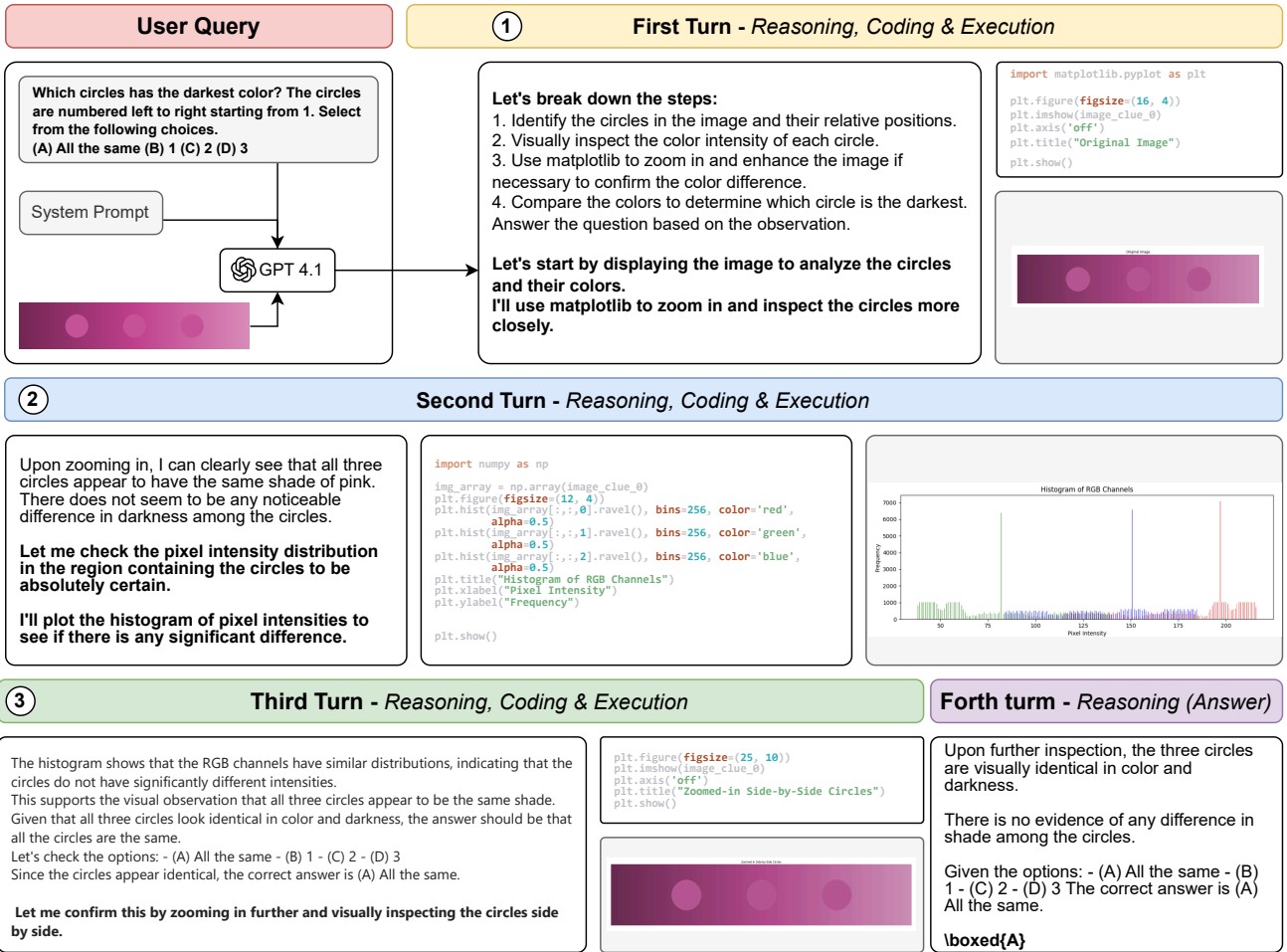

*Figure 17.* **Case Study of Color VQA on TIR-Bench. This task requires** `PyVision`**-Image to analyze the pixels in the image.** This case illustrates how `PyVision`-Image handles a color VQA task, which requires pixel-level image processing. `PyVision`-Image first zooms in on and displays the image, then plots histograms of pixel intensities to examine whether any significant differences exist. The resulting histograms show similar distributions, and based on this pixel-level analysis, `PyVision`-Image arrives at the correct answer.

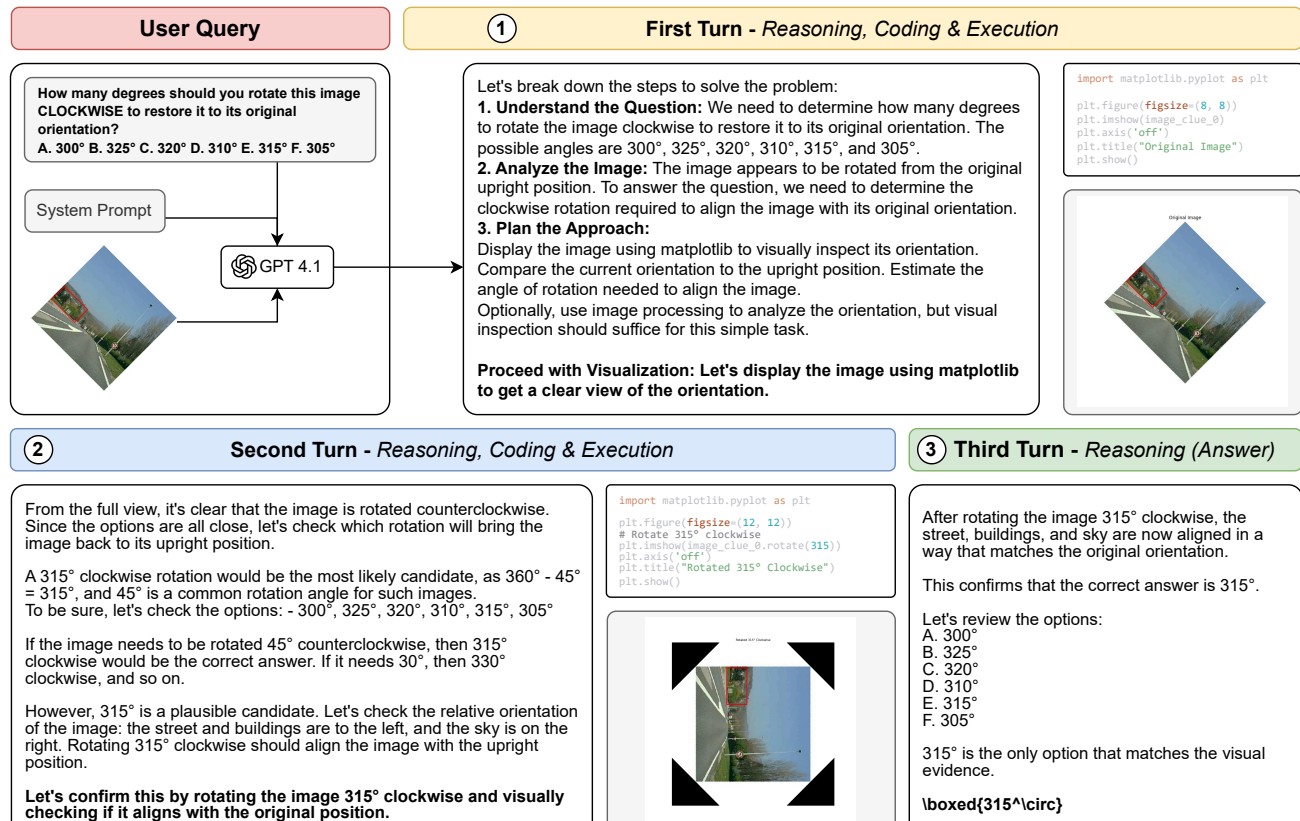

*Figure 18.* **Case Study of Rotation Game on TIR-Bench. This task requires `PyVision`-Image to rotate the images to get the final answers.** This case demonstrates how `PyVision`-Image solves the rotation game task, which requires rotating images to restore them correctly. `PyVision`-Image initially zooms in on and displays the image, then hypothesizes a candidate rotation angle. It subsequently writes Python code to rotate the image and verifies the result. After confirmation, `PyVision`-Image produces the correct final answer.

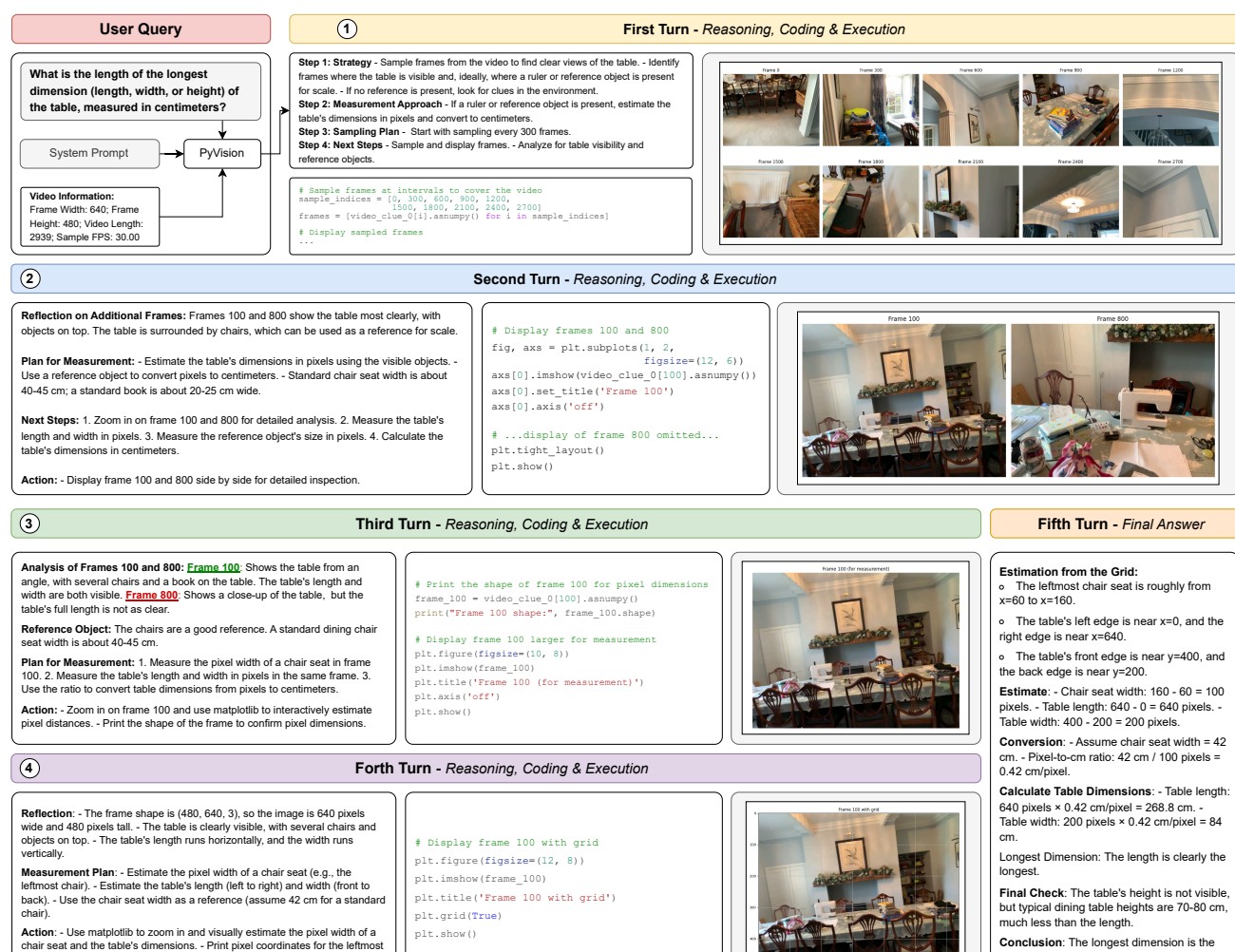

*Figure 19.* **Case Study of Absolute Distance on VSI-Bench. This task requires `PyVision`-Video to first locate the target object and then estimate its distance or physical size.** In this case, `PyVision`-Video estimates the longest dimension of a table from an indoor video. The model first performs uniform frame sampling to identify views where the table and nearby chairs are clearly visible. Using a standard dining chair as a reference object, `PyVision`-Video estimates a pixel-to-centimeter scale and converts the table's pixel span into physical measurements. The table's horizontal length is found to be significantly larger than its width, while the height is not directly observable and is known to be smaller based on typical furniture proportions. Finally, `PyVision`-Video esitimates the longest dimension of the table os 270cm.

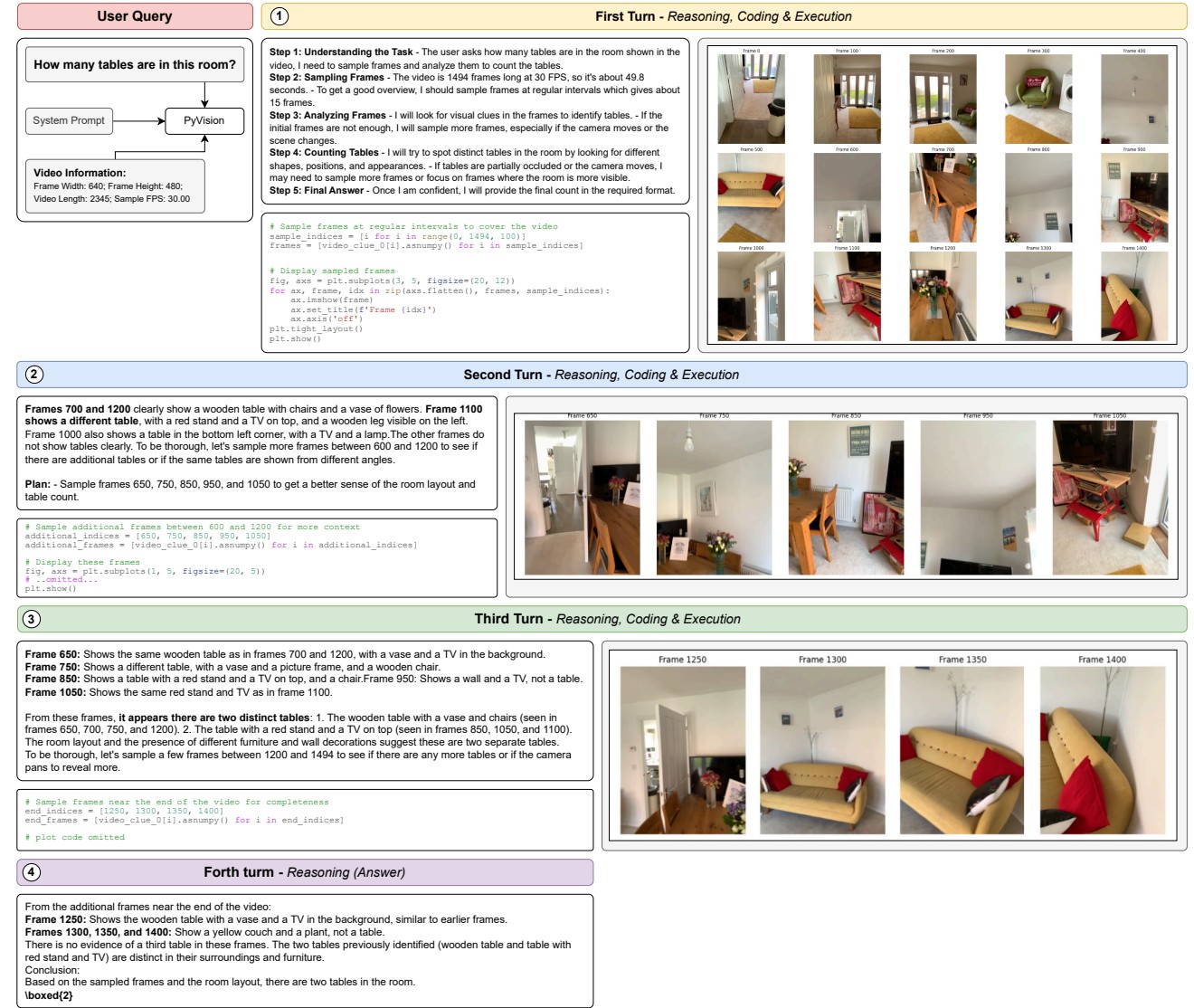

*Figure 20.* **Case Study of Object Counting on VSI-Bench. This task requires `PyVision`-Video to count a specific object in a given video.** In this case, first, `PyVision`-Video uniformly samples 15 frames from the video. Then, it identifies 2 different tables in frame 700 and frame 1100. To see if there are additional tables or if the same tables are shown from different angles, the model samples more frames of the video clip between frame 600 to frame 1200. Finally, based on the constructed context, `PyVision`-Video recognizes two different tables, one wooden table with a vase and chairs and one with a red stand and a TV on top.

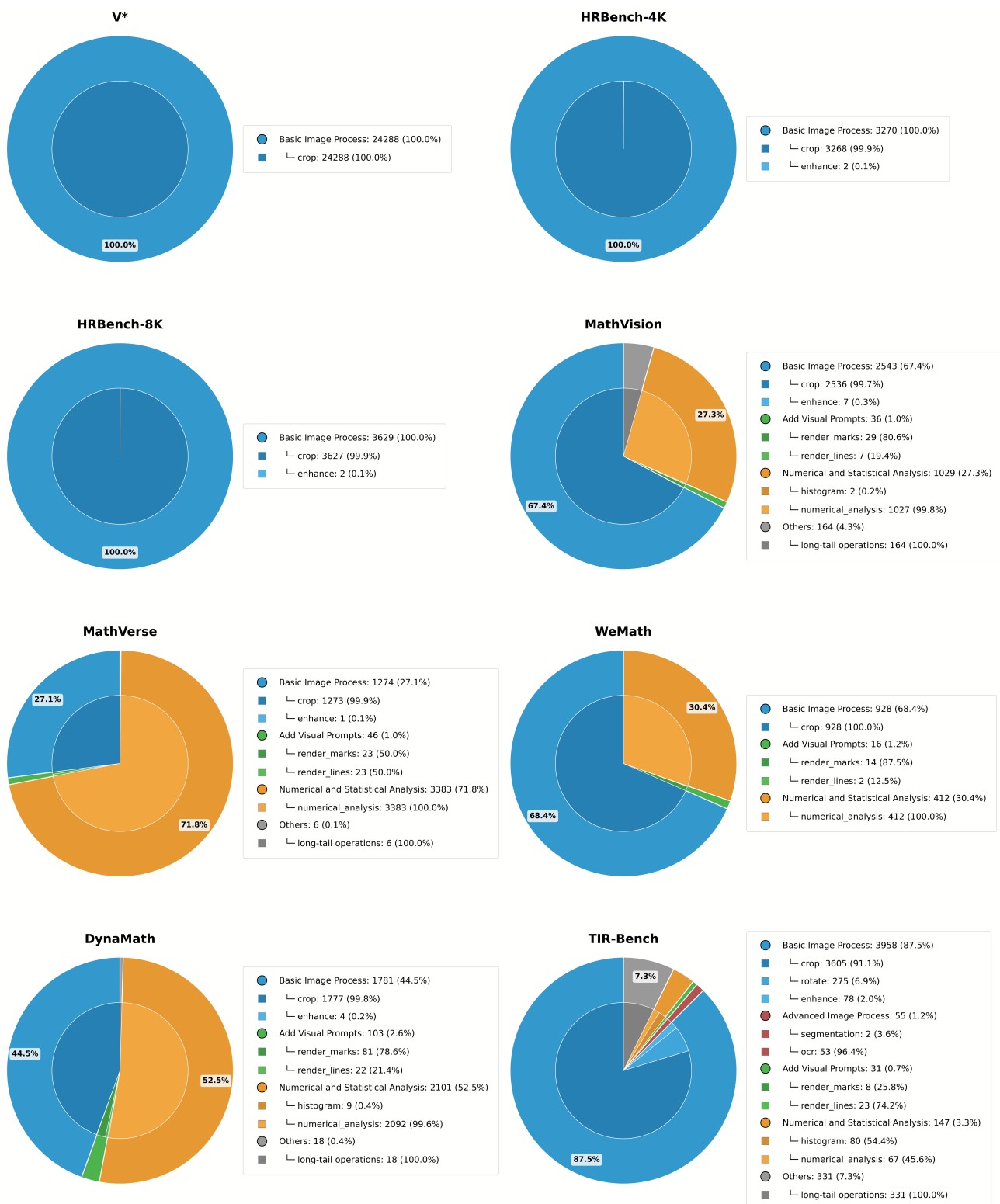

*Figure 21.* **Tooling taxonomy distribution of PyVision-Image on versatile benchmarks.** On visual search tasks, PyVision-Image almost only use *crop* tools. On multi-modal reasoning tasks, PyVision-Image significantly use more *numerical_analysis* tools. On agentic reasoning tasks, i.e., TIR-Bench, PyVision-Image use more diverse tools, including, *segmentation*, *render_marks*, etc, and some long-tail operations, showing dynamic tooling's adaptivability and flexibility.

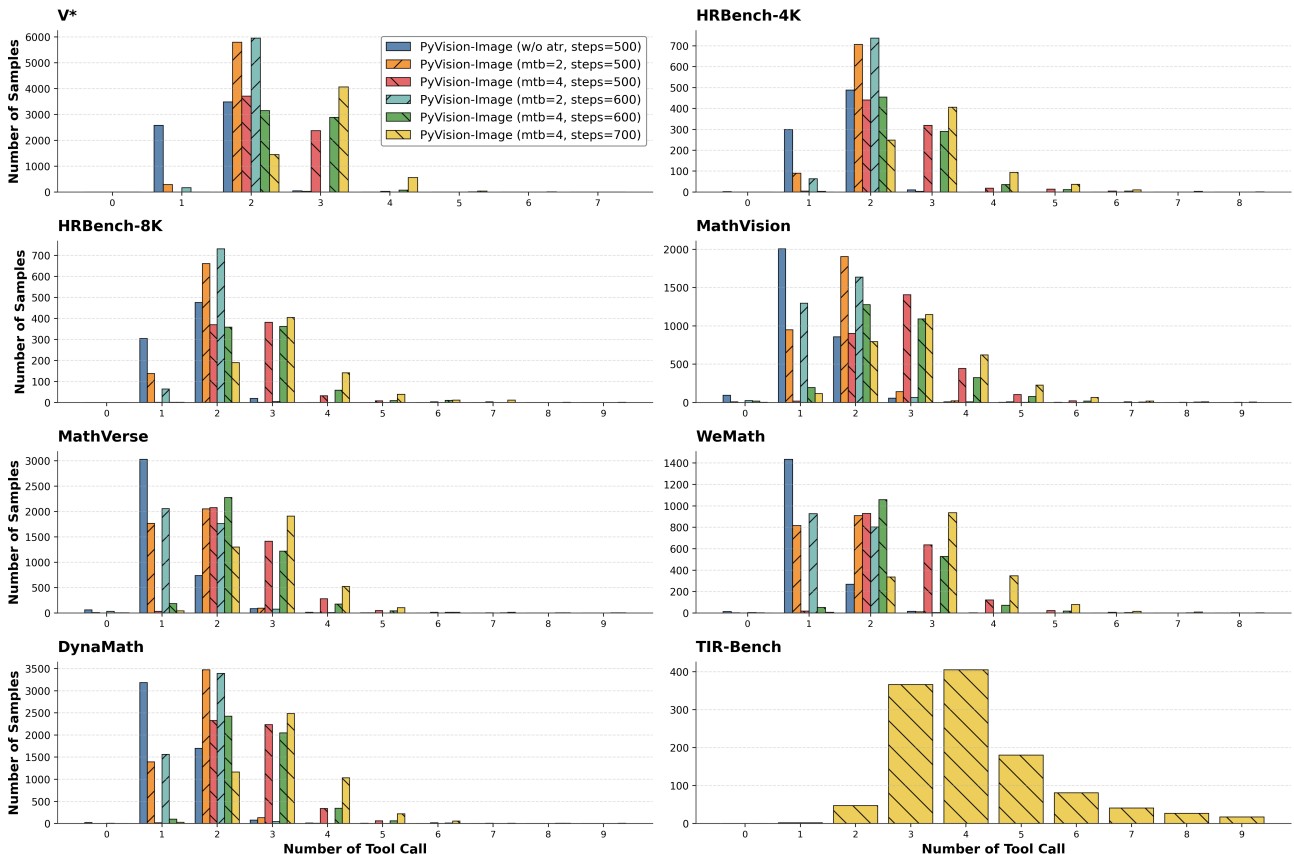

*Figure 22.* **The distribution of tool using number of `PyVision-Image`.** We plot the tool calling number distribution across different benchmarks and models. Models with a larger max turn budget significantly exhibits more tool calling on all benchmarks. On all benchmarks, `PyVision`-Image, trained with maximum turn budget as 4, for 700 steps, use more than 3 turns on most samples, presenting the long-horizon tool using ability.

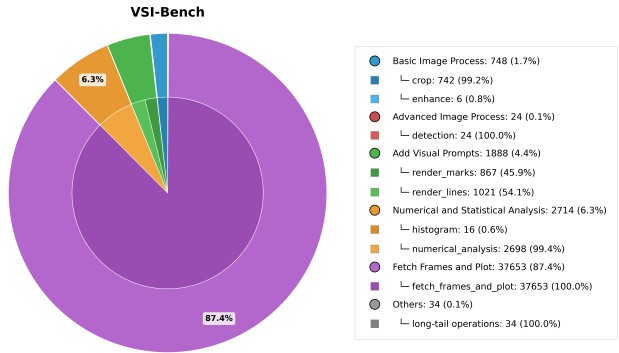

*Figure 23.* **Tooling taxonomy of `PyVision-Video` on VSI-Bench.** We plot the distribution of tool using category distribution of `PyVision`-Video on VSI-Bench. Since the on-demand context construction mechanism, 87.4% tool calling is *fetch_frames_and_plot*. Also, `PyVision`-Video exhibits diverse tool using, indicating the flexibility and adaptivability of dynamic tooling.

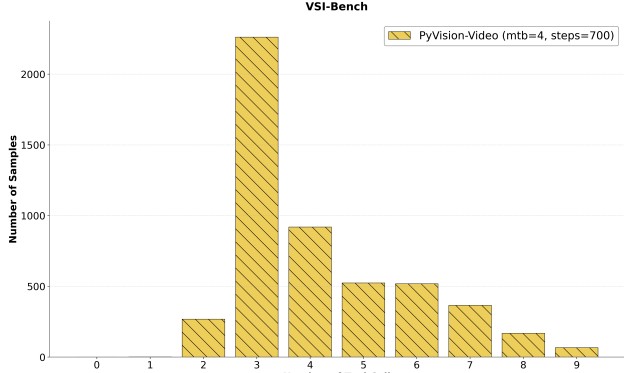

*Figure 24.* **The distribution of tool using number of `PyVision`-Video.** `PyVision`-Video present long-horizon multi-turn tool using ability on VSI-Bench, i.e., most samples are solved with 3 turns and some samples are solved with 9 turns.

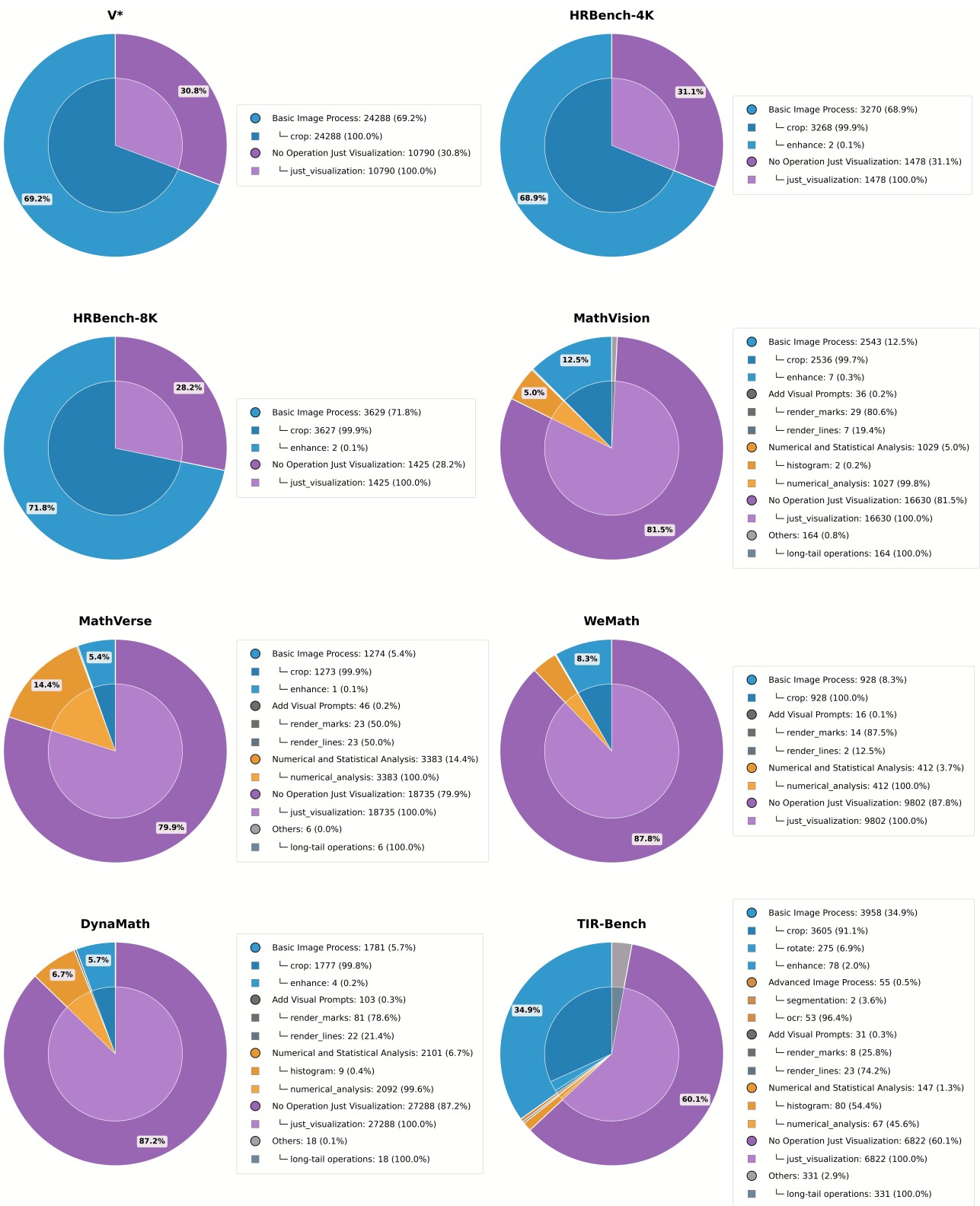

*Figure 25.* **Full tool distribution with no operation.** In this figure, we present the full tooling distribution including the *no operation* as one category, which means the generated Python code just plot the original image without further operation. We find *no operation* accounts for a large portion, indicating that PyVision-Image repeatedly plot the original image to revisit the visual hint.

