# OpenReview forum: "PyVision-RL: Forging Open Agentic Vision Models via RL"
_ICML.cc/2026/Conference — ICML 2026 regular_

### Official Review · Reviewer_n9DV · 2026-03-11

**Soundness:** 3
**Presentation:** 3
**Significance:** 3
**Originality:** 3
**Overall Recommendation:** 5
**Confidence:** 3

**Summary:**

This paper presents PyVision-RL, a reinforcement learning framework that addresses interaction collapse in agentic multimodal model training by combining an oversampling-filtering-ranking rollout strategy with an accumulative tool reward to sustain multi-turn tool use. Applied to both image and video understanding, the framework further introduces on-demand context construction for video reasoning, achieving strong performance while reducing visual token overhead.

**Compliance With Llm Reviewing Policy:**

Affirmed.

**Final Justification:**

My concerns are fully addressed after the rebuttal, and therefore I maintain my positive score.

**Key Questions For Authors:**

See Weaknesses

**Limitations:**

Yes

**Strengths And Weaknesses:**

### Strengths
1. The paper addresses a practically meaningful problem — specifically, the interaction collapse issue that arises during reinforcement learning training of agentic multimodal models.

2. The proposed solution is pragmatic and technically sound. In particular, the use of group-level reward standard deviation as a proxy signal for filtering tasks that are either too difficult or too trivial is an insightful and elegant design.

3. The experimental analysis is thorough and well-organized, providing readers with a clear understanding of the necessity of each component and the underlying reasons for its effectiveness.

### Weaknesses
I do not identify any major weaknesses in this paper. The following are minor suggestions for improvement:

1. Inconsistent training steps across comparisons. I noticed that several figures in the experimental section compare methods that have not been trained for an equal number of RL steps. For instance, in Figure 5, the "w/o std sorting" ablation is only trained up to 300 iterations, and similarly, the baselines in Figures 6 and 7 are not consistently trained to the same number of steps. This inconsistency may compromise the fairness of the comparisons and could potentially misrepresent the relative performance of the methods. I would encourage the authors to either align training budgets across all conditions or provide a clear justification for the discrepancy.

2. Absence of closed-source model comparisons in main results. I would encourage the authors to include comparisons against representative closed-source models in the main results table. Such comparisons would provide readers with a more informative picture of the current gap between open-source and closed-source models on visual reasoning benchmarks.

---

> ### Author Rebuttal · Authors · 2026-03-31
>
> Here we list our responses with the order of the points in Key Questions For Authors:
>
> **(1)** We thank the reviewer for pointing out the inconsistency in training steps across different experimental settings. This discrepancy arises from our limited computational resources, and we apologize for any confusion it may cause.
>
> That said, we believe the current results still provide meaningful support for our conclusions. For example, in Fig. 5, the “w/o std. sorting” setting, even after 500 training steps, shows a clear performance gap compared to our method. Moreover, in Fig. 6 and Fig. 7, notable differences in key metrics appear early in training. In particular, the “w/o standard deviation sorting” setting exhibits a substantially higher positive-sample negative-advantage ratio than the version with standard deviation sorting.
>
> We will clarify this limitation in the revision and, where possible, further align training budgets across settings in future work.
>
> **(2)** Here we add the results of GPT-4o, more results would be added in next version.
>
> | Models          | V*    | HRBench-4K | HRBench-8K | DynaMath | MathVerse | MathVision | WeMath  | TIR-Bench |
> |-----------------|-------|------------|------------|----------|-----------|------------|---------|-----------|
> | Qwen2.5-VL-7B   | 78.5  | 71.6       | 67.9       | 53.3     | 45.6      | 25.6       | 34.6    | 16.0      |
> | PyVision-Image  | 88.7  | 78.1       | 74.3       | 61.6     | 55.8      | 28.7       | 47.7    | 19.8      |
> | GPT-4o          | 66.0  | 59.0       | 55.5       | 63.7     | 41.2      | 30.6       | 42.86   | 17.3      |

---

> > ### Author Rebuttal · Reviewer_n9DV · 2026-03-31
> >
> > My concerns have been adequately addressed. Therefore, I keep my positive score.

---

### Official Review · Reviewer_cRgK · 2026-03-12

**Soundness:** 3
**Presentation:** 3
**Significance:** 3
**Originality:** 3
**Overall Recommendation:** 4
**Confidence:** 3

**Summary:**

This paper introduces PyVision-RL, an agentic reinforcement learning (RL) framework designed for open-weight multimodal models. The core philosophy treats Python as a primitive tool to provide unified support for both image and video reasoning. Specifically, PyVision-Video eschews direct injection of entire videos into the model context; instead, it maintains the video within a runtime environment, allowing the model to perform on-demand sampling and frame plotting during inference. This approach effectively constructs the necessary context while significantly reducing visual token consumption. Regarding training, the paper proposes two pivotal mechanisms: (1) accumulative tool rewards, which grant additional credit for tool calls upon correct answers to mitigate interaction collapse; and (2) an oversampling–filtering–ranking strategy for rollout selection. Furthermore, the framework stabilizes training by removing standard deviation normalization in GRPO. Experimental results demonstrate that PyVision-Image achieves superior performance across multiple image benchmarks, while PyVision-Video offers a compelling accuracy–efficiency trade-off on VSI-Bench.

**Compliance With Llm Reviewing Policy:**

Affirmed.

**Final Justification:**

Thank you for the detailed rebuttal. The clarifications regarding data sources and interaction budget analysis are helpful and improve the clarity of the paper. However, my main concerns remain, particularly regarding the lack of more rigorous controlled comparisons (e.g., fixed-budget settings and ablations on frame sampling strategies). Therefore, I decide to keep my original score.

**Key Questions For Authors:**

1.Can the authors provide explicit train/eval disjointness and contamination checks?
In particular, are there any sample-level, template-level, or source-level overlaps between the training data and evaluation benchmarks?

2.Can the authors provide cost-aware or utility-normalized analyses?
For example, under fixed tool-call budgets, fixed wall-clock budgets, or fixed token budgets, does the method still outperform baselines?

3.Can the authors isolate the effect of on-demand context construction more rigorously?
I would like to see a direct comparison between on-demand frame retrieval and fixed uniform frame sampling under the same base model, training data, and inference budget.

**Limitations:**

The current Impact Statement briefly mentions the security risks of Python tool usage, but a more comprehensive discussion is needed. I suggest the authors address the following:

1.Narrow Video Evaluation: The limited coverage of video benchmarks and potential issues with generalizability.

2.Redundant Tool Use: Whether the accumulative tool reward inadvertently encourages unnecessary or non-essential tool calls.

3.Sandbox Robustness: The underlying security assumptions and potential failure modes of the execution environment.

**Strengths And Weaknesses:**

Strengths:

1.Significance of the Research Problem: The paper addresses a critical and timely research direction: enhancing RL training stability and tool-use capabilities for open-weight multimodal agents.

2.Unified Methodological Framework: The framework is conceptually elegant, leveraging Python as a versatile tool to provide a unified approach for both image and video reasoning.

3.Sound Technical Design: The motivations behind the technical components are clear; specifically, the introduction of accumulative tool rewards and rollout filtering/ranking effectively targets the mitigation of interaction collapse.

4.Robust Experimental Results: The model achieves competitive performance on image benchmarks and demonstrates a favorable accuracy–efficiency trade-off in video-based tasks.

5.Insightful Visualizations and Ablations: The inclusion of training curves, component analyses, and qualitative case studies provides strong empirical support for the paper’s primary claims.

Weaknesses

1.Concerns Regarding Reward Design: There are lingering questions about the reward mechanism. While the current design may incentivize a higher frequency of tool calls, the paper does not sufficiently demonstrate that this translates to more effective or optimized tool utilization.

2.Constrained Scope of Video Experiments: The evaluation for video tasks is relatively narrow, focusing primarily on a single benchmark. This leaves the framework's generalizability across a broader range of video-based tasks under-supported.

3.Rigor of Comparative Analysis: The fairness of the comparisons could be strengthened. Specifically, the paper lacks explicit data contamination checks and a more rigorous comparison under consistent test-time budgets.

4.Expository Clarity: There is room for improvement in the writing. Certain technical descriptions and mathematical notations require more precise definitions to ensure full clarity.

5.Insufficient Discussion of Limitations: The manuscript does not adequately address potential limitations, such as safety considerations, deployment costs, inference latency, and other inherent risks.

---

> ### Author Rebuttal · Authors · 2026-03-31
>
> Here we list our responses with the order of the points in Key Questions For Authors:
>
> **(1)** We thank the reviewer for the question regarding the training data.
>
> We have provided detailed distributions of both SFT and RL data in Fig. 11 and Fig. 12. All datasets and benchmarks used in this work are sourced from the open-source community, and their effectiveness has been validated in numerous prior studies.
>
>
> **(2)** We acknowledge that it is difficult to strictly control the inference budget across different models, which limits our ability to perform perfectly matched comparisons. Nevertheless, we provide empirical analysis to support our findings. In Fig. 5, we train PyVision-Image under different interaction budgets, and in Fig. 22, we observe that models trained with higher interaction budgets tend to complete tasks using more interaction turns.
>
> Based on this, we implement a coarse-grained control of inference budget by varying the interaction budget during training, and demonstrate interaction scaling behavior in Fig. 5.
>
>
> **(3)** We agree that ablation studies on this component would provide additional insights. However, due to limited computational resources and the high inference cost associated with incorporating uniform frame sampling into the agent scaffold, we were unable to include this ablation in the current work.
>
> Nevertheless, we provide indirect evidence through comparison with existing methods. As shown in Fig. 4, baseline models such as Qwen, Video-R1, and SpaceR all employ uniform frame sampling. In contrast, PyVision-Video achieves the best trade-off between performance and visual token efficiency.

---

> > ### Author Rebuttal · Reviewer_cRgK · 2026-04-03
> >
> > Thank you for the detailed rebuttal. The clarifications regarding data sources and interaction budget analysis are helpful and improve the clarity of the paper. However, my main concerns remain, particularly regarding the lack of more rigorous controlled comparisons (e.g., fixed-budget settings and ablations on frame sampling strategies). Therefore, I decide to keep my original score.

---

### Official Review · Reviewer_YzCx · 2026-03-13

**Soundness:** 3
**Presentation:** 3
**Significance:** 2
**Originality:** 2
**Overall Recommendation:** 4
**Confidence:** 4

**Summary:**

This paper introduces PyVision-RL, an RL framework that trains open-weight multimodal models to use Python as a tool during inference. The main technical contributions are an oversampling-filtering-ranking strategy and an accumulative tool reward to prevent interaction collapse during RL training. Two models are produced: PyVision-Image and PyVision-Video, where the latter uses on-demand frame sampling. Results are reported on visual search, multimodal reasoning, and video benchmarks.

**Compliance With Llm Reviewing Policy:**

Affirmed.

**Key Questions For Authors:**

1. On which task types does tool usage actually help, and on which does it not? A per-category analysis of tool call count vs. accuracy gain would be very informative.
2. How sensitive is performance to the 0.1 tool reward coefficient? What happens at 0.0, 0.05, 0.2, 0.5?
3. Does interaction collapse happen with other base models (InternVL, LLaVA)?
4. What fraction of tool calls in the final model are actually necessary? What's the accuracy when you cap tool calls at 1 or 2?
5. On failed examples, is the bottleneck perception, code generation, or reasoning?

**Limitations:**

No dedicated limitations section. The impact statement briefly notes that Python-based tooling could access the host file system and cause damage, but no technical mitigation is proposed.

**Strengths And Weaknesses:**

**Strengths:**

1. Interaction collapse is a real problem and the paper does a good job diagnosing it. The oversampling-filtering-ranking strategy is simple and the ablations (Fig. 5, 6, 7) back it up.

2. The on-demand frame sampling for video is probably the most useful idea here. Letting the agent decide which frames to load cuts token usage by ~9x and actually improves accuracy (Fig. 4).

3. The evaluation is fairly broad — visual search, multimodal reasoning, agentic tasks, and video spatial reasoning — and results are consistent under one unified framework.

4. Training dynamics in Fig. 3 show stable optimization with growing tool usage, which is reassuring.

**Weaknesses:**

1. The accumulative tool reward always pushes toward more tool calls with no mechanism to decide when tool use actually helps. Recent work on multimodal reasoning shows models can learn to selectively skip structured reasoning when it's unnecessary. PyVision doesn't have anything like this — it just rewards more interaction regardless. This risks over-reasoning on easy queries.

2. Related to the above: the paper never asks "when does multi-turn tool use help vs. hurt?" No per-task or per-difficulty breakdown of tool usage patterns is provided. Knowing which tasks benefit from 1 vs. 3 vs. 5 tool calls would make this a much stronger contribution.

3. Only aggregate accuracy is reported. There's no error taxonomy, no failure analysis, no skill-level breakdown. You can't tell whether the gains come from better perception or better reasoning.

4. Only one base model (Qwen2.5-VL-7B). Interaction collapse might be specific to this architecture or scale — hard to know without testing on at least one other model.

5. The tool reward coefficient (0.1) is never ablated. This controls the core accuracy-interaction tradeoff and deserves more attention.

6. SFT data is GPT-4.1 synthetic with no quality analysis. Building on noisy demonstrations without checking their reliability is risky.

7. Python execution security is flagged in the impact statement but no technical solution is proposed. For a system built around arbitrary code execution, this matters.

---

> ### Author Rebuttal · Authors · 2026-03-31
>
> Here we list our responses with the order of the points in Key Questions For Authors:
>
> **(1)** Here we provide a detailed analysis of average tool-call count and accuracy across different task categories under varying maximum turn budgets (MTB) during training. Overall, we observe a consistent positive correlation between increased tool usage and improved performance.
>
>
> |MTB| V*                |       | HRB-4k       |       | HRB-8k       |       | DynaMath         |       | MathVerse        |       | MathVision       |       | WeMath           |       |
> |---|-------------------|-------|------------------|-------|------------------|-------|------------------|-------|------------------|-------|------------------|-------|------------------|-------|
> || count   | acc   | count  | acc   | count  | acc   | count  | acc   | count  | acc   | count  | acc   | count  | acc   |
> | 2| 1.97              | 84.47 | 1.92             | 76.38 | 1.92             | 71.37 | 1.69             | 60.02 | 1.48             | 52.66 | 1.59             | 28.67 | 1.46             | 44.38 |
> |4| 2.50              | 86.24 | 2.51             | 77.72 | 2.68             | 72.22 | 2.68             | 61.58 | 2.59             | 57.31 | 2.71             | 28.66 | 2.49             | 47.71 |
>
>
> In particular, except for MathVision, higher tool-call counts generally lead to better task performance across categories. This suggests that allowing more interaction steps enables the model to more effectively leverage external tools for problem solving.
>
> **(2)** We thank the reviewer for the question regarding the tool reward coefficient.
>
> We have included the setting with a tool reward coefficient of 0.0 (i.e., without accumulative tool reward) in Fig. 5. The results show that after 500 RL steps, our method achieves a clear performance improvement over the variant without accumulative tool reward.
> For additional coefficients (0.05, 0.2, and 0.5), we agree that a more comprehensive comparison would strengthen the analysis. However, due to limited computational resources, we were unable to include these experiments in the current paper.
> We will explicitly acknowledge this limitation in the revision and consider expanding this study in future work.
>
> **(3)** We thank the reviewer for the insightful question regarding interaction collapse.
>
> We believe that interaction collapse is closely related to the training stage and data composition of the base model, particularly whether it has been pretrained or mid-trained with long chain-of-thought (CoT) data. Prior work [1] shows that, under the same training data and strategy, LLaMA exhibits decreasing response length with more RL steps, while Qwen shows the opposite trend. However, after additional mid-training on long and complex reasoning data, LLaMA also demonstrates increasing response length during RL.
>
> This observation is consistent with our findings. We hypothesize that incorporating more long-horizon reasoning and tool-use data during pre-training and mid-training can mitigate or prevent interaction collapse.
>
> **(4)** As observed in **(1)**, models trained with a lower maximum turn budget tend to use fewer tools during inference (fewer than 2 on average), which corresponds to worse overall performance. Across 6 benchmarks, increased tool usage consistently leads to substantial performance improvements.
>
> **(5)** Following the reviewer’s guidance, we first defined three categories of failure:
> - Perception: The model fails to correctly interpret or extract visual information from the image.
> - Code generation: The model produces incorrect or invalid code, leading to an incorrect final answer.
> - Reasoning: The model’s logical deduction or multi-step reasoning is flawed.
>
> |                  | V*           | HRB-4k   | HRB-8k   | DynaMath      | MathVerse     | MathVision    | WeMath        |
> |------------------|--------------|--------------|--------------|---------------|---------------|---------------|---------------|
> | perception       | 607 (87.84%) | 113 (70.62%) | 141 (68.45%) | 658 (33.38%)  | 399 (22.85%)  | 346 (15.94%)  | 112 (23.38%)  |
> | code generation  | 0 ( 0.00%)   | 0 ( 0.00%)   | 0 ( 0.00%)   | 0 ( 0.00%)    | 0 ( 0.00%)    | 1 ( 0.05%)    | 0 ( 0.00%)    |
> | reasoning        | 84 (12.16%)  | 47 (29.38%)  | 65 (31.55%)  | 1313 (66.62%) | 1347 (77.15%) | 1791 (82.53%) | 332 (69.31%)  |
>
>
> We then used GPT-5.4 to analyze the failure type of samples across the eight benchmarks. Our findings are as follows:
>
> (a) Only a few failures are caused by generated invalid or non-executable code snippets.
>  (b) In visual search benchmarks, perception is the main bottleneck for most failures.
>  (c) In multi-modal math benchmarks, reasoning is the dominant bottleneck.
>
>
> **[1]** Cognitive Behaviors that Enable Self-Improving Reasoners, or, Four Habits of Highly Effective STaRs

---

> > ### Author Rebuttal · Reviewer_YzCx · 2026-04-04
> >
> > Thanks. I keep my positive score.

---

### Official Review · Reviewer_aU56 · 2026-03-16

**Soundness:** 4
**Presentation:** 4
**Significance:** 4
**Originality:** 4
**Overall Recommendation:** 6
**Confidence:** 4

**Summary:**

This paper presents PyVision-RL, a framework for training open-weight vision agents with RL while mitigating interaction collapse. The key ideas are a simple Python-based tool interface, an oversampling–filtering–ranking strategy to stabilize rollouts, and an accumulative tool reward to encourage sustained multi-step tool use. Built on this framework, the authors develop both PyVision-Image and PyVision-Video, with the video model further using on-demand context construction to retrieve relevant frames during reasoning. Experiments show strong image results and an appealing accuracy-efficiency trade-off on video benchmarks.

**Compliance With Llm Reviewing Policy:**

Affirmed.

**Final Justification:**

My concerns have been adequately addressed. I will raise my score to Strong Accept

**Key Questions For Authors:**

1. The accumulative reward is proportional to the number of tool calls for correct rollouts. Did you test alternatives that reward useful tool calls rather than simply more tool calls, and how sensitive are results to the coefficient 0.1?

2. For the oversampling–filtering–ranking strategy, how much gain comes from each part individually? The current paper supports the combined effect, but a cleaner decomposition would strengthen the claim.

**Limitations:**

yes

**Strengths And Weaknesses:**

Strengths:

This paper addresses a real RL failure mode for agentic MLLMs: interaction collapse, i.e., tool use drops during training. The proposed accumulative tool reward is well aligned with this problem, and the ablation is convincing: removing it reduces tool usage, hurts later-stage performance after about 500 training steps, and weakens long-horizon reasoning.

Weaknesses:

1. The tool reward is proportional to the number of tool calls for correct rollouts, so it may partially reward “longer successful trajectories” rather than genuinely better tool selection or more efficient reasoning.

2. You mention the use of the sandbox, but the execution boundary remains insufficiently specified. In the appendix, the system prompt states that “The Python code will be executed by an external sandbox,” which indicates that you do deploy an external sandboxing mechanism. However, the paper does not further clarify the actual restrictions enforced by this sandbox, such as whether file-system access is disabled, whether network access is blocked, whether system calls are restricted, or whether CPU, memory, and execution-time limits are imposed.
Moreover, the impact statement explicitly acknowledges that Python, as a primitive tool, “may access the host file system and makes damage.” This suggests that you are aware of the associated risks. I think it's better to provide sufficient details on the concrete protection measures.

3. The on-demand frame retrieval design is efficient, but it may occasionally miss short-lived or scattered visual evidence that would be easier to capture with fuller video coverage. This does not undermine the strong results, but you could discuss this trade-off more explicitly.

4. The paper could also provide a bit more detail on robustness to code execution failures. Since the model generates Python code during reasoning, issues such as execution errors or timeouts are likely to arise in practice. It would be helpful to clarify how these cases are handled and whether excluding failed trajectories has any effect on training.

---

> ### Author Rebuttal · Authors · 2026-03-31
>
> Here we list our responses with the order of the points in Key Questions For Authors:
>
> **(1)** We thank the reviewer for raising the question regarding the evaluation of tool-use usefulness and the choice of tool reward coefficients.
>
> Regarding usefulness, we note that in general visual QA settings, it is inherently challenging to rigorously define and verify whether a tool call is “useful.” In our framework, seemingly “useless” tool calls may in fact reflect the model’s exploration of the task space. Therefore, for simplicity and clarity, we do not explicitly distinguish between useful and non-useful tool usage.
>
> As for the tool reward coefficients (e.g. 0.05, 0.2, and 0.5), we acknowledge that a more comprehensive comparison would be valuable. However, due to limited computational resources, we were unable to include these additional experiments in the current submission.
> We will clarify these design choices and limitations in the revision and consider a more thorough exploration of reward coefficients in future work.
>
>
> **(2)** In the ablation section, we isolate and evaluate the impact of the ranking strategy. Specifically, Fig. 5 and Table 3 present the effect of ranking on final performance in an independent manner.
>
> For filtering, we emphasize that its primary role is to stabilize the training process rather than directly improve final performance. In practice, removing the filtering mechanism leads to instability and even failure of the training pipeline. For example, one of our filtering policies removes samples containing more than 50 images. Without this constraint, certain rollouts exceed the context window of Qwen2.5-VL, which results in inference server crashes and termination of the training process.

---

> > ### Author Rebuttal · Reviewer_aU56 · 2026-04-02
> >
> > My concerns have been adequately addressed. I will raise my score to Strong Accept. Very nice work!

---

### Decision · Program_Chairs · 2026-04-30

**Decision:**

Accept (regular)

**Comment:**

The paper presents a framework for mitigating tool-use collapse in agentic frameworks, where it is frequently observed that the frequency of model interactions with tools to solve a given problem diminishes during reinforcement learning (RL) training. Two remedies are proposed in the paper utilizing a Python-based tool use interface: i) incorporating rewards in RL that incentivizes tool interaction, and ii) employing an oversampling-filtering-ranking-based GRPO RL training approach. This approach involves oversampling rollouts, filtering to eliminate broken interactions, and ranking them based on pseudo-difficulty captured by reward variance. The paper asserts that this method facilitates stable and effective training. Experiments are conducted on visual search, multimodal, and agentic reasoning tasks, demonstrating substantial performance enhancements. Specifically PyVision-Video for loading entire videos into Python run-time context and on-demand frame-sampling is an interesting idea.

The paper received generally positive reviews. Reviewers acknowledged the significance and timeliness of the problem being addressed and the strong results reported. However, concerns were raised regarding the balance between episode length and interaction frequency, its effectiveness, the absence of failure analysis, and limited comparisons on video-based tasks. Authors provided additional experimental results and details that addressed some of these concerns, while acknowledging that analyzing the precise utility of the approach under tool-call budgets over varied models is a difficult problem.

AC had an independent reading of the paper and concurs with the reviewers that the paper explores an important problem in agentic learning for multimodal data. The technical contributions are timely, and the empirical results are extensive. The paper could benefit from a deeper exploration of the underlying causes of reduced tool use, possibly through specific examples, as suggested by the reviewers. Overall, the paper makes a solid contribution, and AC recommends acceptance.